# Metafiber transforming arbitrarily structured light

Chenhao Li[1], Torsten Wieduwilt [2], Fedja J. Wendisch[1], Andrés Márquez [3,4], Leonardo de S. Menezes[1,5], Stefan A. Maier [1,6,7] ✉, Markus A. Schmidt [2,8,9] ✉ & Haoran Ren [6] ✉

Structured light has proven useful for numerous photonic applications. However, the current use of structured light in optical fiber science and technology is severely limited by mode mixing or by the lack of optical elements that can be integrated onto fiber end-faces for wavefront engineering, and hence generation of structured light is still handled outside the fiber via bulky optics in free space. We report a metafiber platform capable of creating arbitrarily structured light on the hybrid-order Poincaré sphere. Polymeric metasurfaces, with unleashed height degree of freedom and a greatly expanded 3D meta-atom library, were 3D laser nanoprinted and interfaced with polarization-maintaining single-mode fibers. Multiple metasurfaces were interfaced on the fiber end-faces, transforming the fiber output into different structured-light fields, including cylindrical vector beams, circularly polarized vortex beams, and arbitrary vector field. Our work provides a paradigm for advancing optical fiber science and technology towards fiber-integrated light shaping, which may find important applications in fiber communications, fiber lasers and sensors, endoscopic imaging, fiber lithography, and lab-on-fiber technology.

The most common type of structured light represents singular optical beams carrying spatially inhomogeneous polarizations on a helical wavefront with orbital angular momentum (OAM), which can be generalized as arbitrary spin–orbit coupling states on the hybrid-order Poincaré sphere (HOPS)[1]. Structured light fields have proven useful for sub-diffraction focusing[2,3], optical trapping[4–6], multimode imaging[7–9] and holography[10–13], free-space and fiber communications[14,15], nonlinear light conversion[16], chiral sensing[17,18], and quantum information processing[19–22]. Conventional structured-light generation typically requires multiple cascaded phase and wave plates such as bulky spatial light modulators, imposing major challenges for any practical use and applications. Metasurfaces composed of subwavelength meta-atoms

(the unit cell structures of metasurfaces) have recently transformed photonic design[23–26], opening the possibility of using ultrathin photonic devices to generate[27–30], detect[31–34], and manipulate structured light[29,35–37]. Plasmonic and dielectric meta-atoms with phase shifts from 0 to 2π of scattered light have been developed to realize phase singularities carrying the OAM of light[25,35,36]. Meanwhile, anisotropic meta-atoms acting as subwavelength waveplates have been developed to imprint polarization singularities in cylindrical vector beams[24] and perfect Poincaré beams[28].

Implementing structured light on optical fibers could be crucial for widespread fiber applications ranging from fiber communications[15], fiber lasers[38], and fiber sensors[39] to endoscopic imaging[40], fiber

[1]Chair in Hybrid Nanosystems, Nanoinstitute Munich, Faculty of Physics, Ludwig Maximilian University of Munich, 80539 Munich, Germany. [2]Leibniz Institute of Photonic Technology, 07745 Jena, Germany. [3]I.U. Física Aplicada a las Ciencias y las Tecnologías, Universidad de Alicante, P.O. Box 99, 03080 Alicante, Spain. [4]Dpto. de Física, Ing. de Sistemas y Teoría de la Señal, Universidad de Alicante, P.O. Box 99, 03080 Alicante, Spain. [5]Departamento de Física, Universidade Federal de Pernambuco, 50670-901 Recife-PE, Brazil. [6]School of Physics and Astronomy, Faculty of Science, Monash University, Melbourne, Victoria 3800, Australia. [7]Department of Physics, Imperial College London, London SW7 2AZ, UK. [8]Abbe Center of Photonics and Faculty of Physics, FSU Jena, 07745 Jena, Germany. [9]Otto Schott Institute of Material Research, FSU Jena, 07745 Jena, Germany. ✉e-mail: stefan.maier@monash.edu; markus.schmidt@leibniz-ipht.de; haoran.ren@monash.edu

lithography[41], and lab-on-fiber technology[42,43]. However, the practical use of structured light from optical fibers is severely hampered by modal crosstalk and polarization mixing or by the lack of optical elements that can be integrated onto fiber end-faces for complex wavefront manipulation. To date, the generation of structured light is still handled in the majority of cases outside the fiber via bulky optics in free space, which can hinder the deployment of structured light for fiber science and technology and partially nullifies the advantages of optical fiber such as flexible light guidance. Here, promising approaches implementing meta-structures on fiber end faces have been used to demonstrate Bessel beam converters[44], fiber couplers[45], polarization controllers and waveplates[46]. Although electron-[42,47–49] and ion-beam lithography[50,51], nanoimprinting[52] and hydrofluoric chemical-etching techniques[53] have been proposed to implement metasurfaces on the fiber end-faces, these fabrication methods suffer from either a complicated manufacturing process or difficulties in interfacing arbitrary 3D nanostructures for efficient wavefront engineering. 3D laser nanoprinting, based on two-photon polymerization, has been introduced to interface 3D micro-optics on fiber tips[54–59]. Recently, 3D laser-nanoprinted lenses with high numerical aperture[57], achromatic focusing[60], multifocus generation[61], and inverse-design optimization[41] have been integrated on fiber end-faces to improve fiber functionalities and applications. Nevertheless, it remains elusive to realize arbitrarily structured light directly on optical fibers.

Here, we show the design, 3D laser nanoprinting, and characterization of structured light-generating metafibers (SLGMs), capable of on-fiber transforming arbitrarily structured light fields on the HOPS (Fig. 1). For sample implementation and polarization manipulation, we experimentally use a commercial polarization-maintaining single-mode fiber (PM-SMF, PM1550-XP, Thorlabs). Details on the fiber used can be found in Supplementary Note 1. To allow the output of the

PM-SMF to be freely expanded to fully cover the area of a 3D metasurface with a diameter of around 100 μm without the use of additional optical components (e.g., GRIN lenses), we first use laser to print a hollow tower structure (height of 550 μm) onto the fiber end-face (Fig. 1a). The impact of the tower structure on the fiber beam output is studied in Supplementary Note 2. Note that the tower and the metasurface were printed together in one run, i.e., they thus are made of the same material (here IP-L polymer). To prove our concept, we demonstrate several different SLGMs with various structured light outputs including the radial (SLGM-1) and azimuthal (SLGM-2) polarizations, circularly polarized vortex beams with topological charges of −1 (SLGM-3) and −3 (SLGM-4), as well as an arbitrary vector field on the HOPS with spatially variant localized elliptical polarizations (SLGM-5) (Fig. 1b). Note that in this work, we do not include the phase profile of a lens in our metasurface design to collimate the beam, since our priority is to demonstrate the generation of complex beams using a fiber-integrated system. We should mention that a lens profile can be implemented onto our metasurface design when a collimated fiber output is required, which is empowered by the simultaneous and independent control of both the polarization and phase responses in our metasurface. We show that 3D anisotropic meta-atoms with unleashed height degree of freedom offer a greatly expanded 3D meta-atom library, allowing the independent, complete, and precise polarization and phase control at the level of a single meta-atom, paving the way for the use of a single metasurface to create arbitrarily structured light on the fiber end-face.

## Results
### Design principle
An arbitrarily structured light field on the HOPS can be defined mathematically as a superposition of left- and right-handed circular

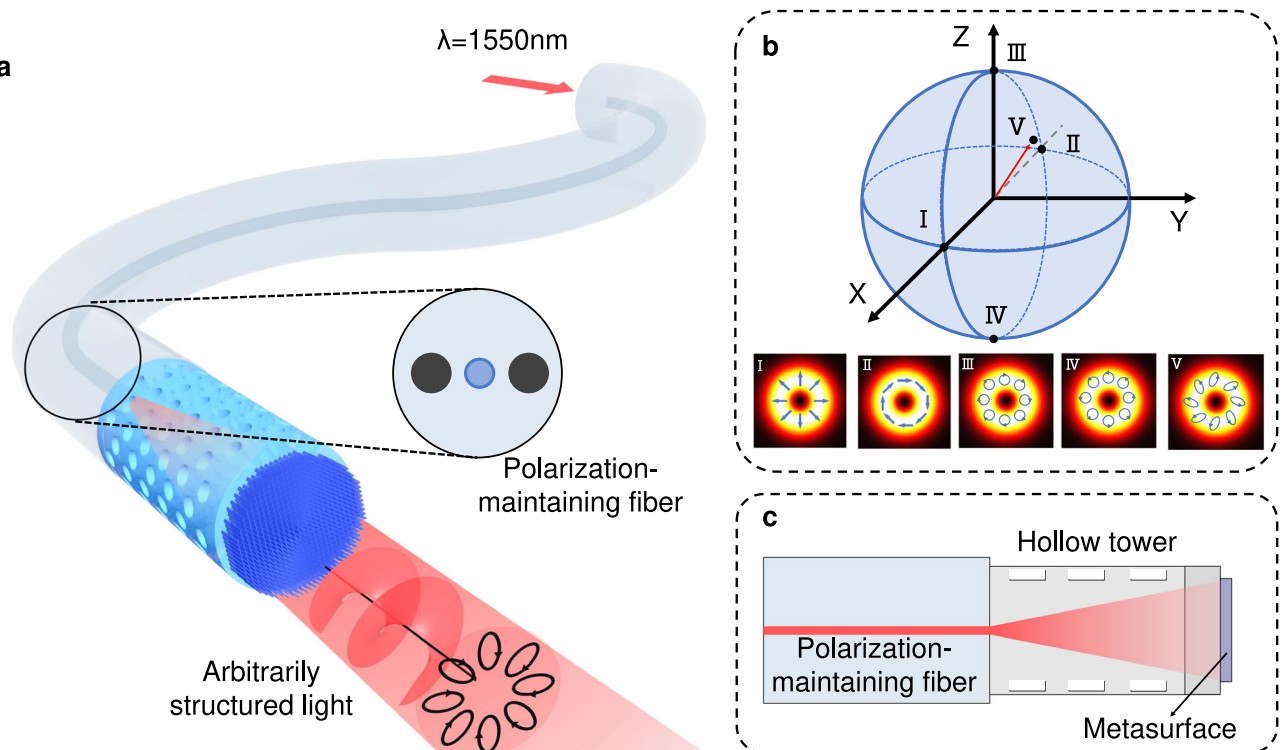

**Fig. 1 | Principle of generating arbitrarily structured light on metafibers.**
**a** Schematic representation of a polarization-maintaining fiber interfaced with a 3D laser-nanoprinted metasurface on top of a hollow tower structure. The metafiber output features an arbitrarily structured light field. **b** The realized structured light fields on a HOPS carry spatially variant polarization distributions, with some examples indicated by the states I to V. The arrows refer to the local polarization of the electric field. **c** Schematic of the side view of the metafiber.

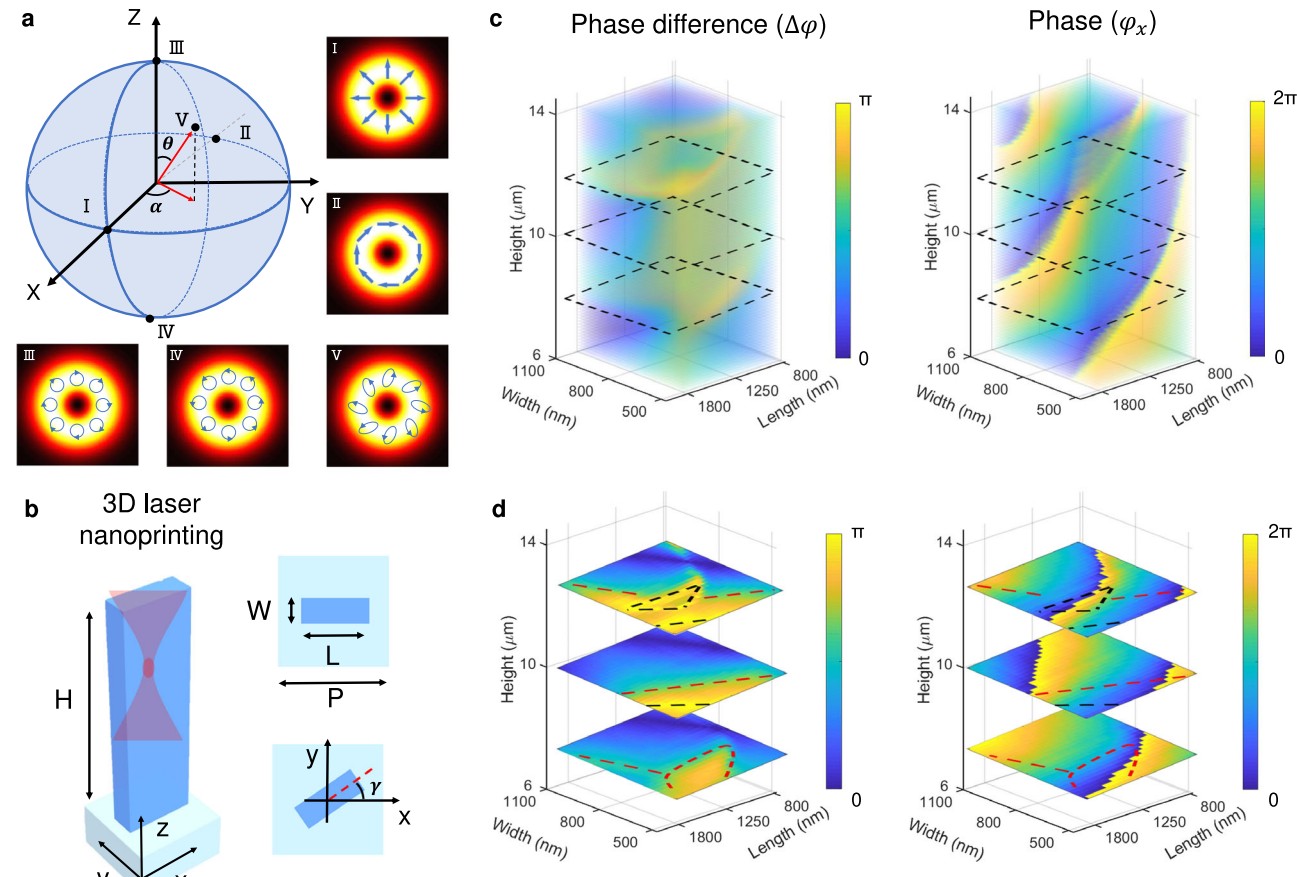

**Fig. 2 | Design of a 3D nanopillar used to implement metasurfaces for generating arbitrarily structured light beams. a** Showcase of five representative structured light fields on the first-order HOPS, defined by different angles of $\theta$ and $\alpha$. **b** Schematic of a 3D laser nanoprinted nanopillar waveguide in a polymer matrix ($H$: height, $W$: width, $L$: length, $\gamma$: the in-plane rotation angle of the nanopillar with respect to x-axis). **c** Simulated 3D meta-atom library consists of the phase difference between the x- and y-linear transverse modes (left) and the propagation phase of the light polarized along the x-direction (right). **d** Three exemplary data planes in the meta-atom library are highlighted, with heights of $H = 8, 10, 12\ \mu m$, respectively. Red and black dashed lines mark the 3D nanopillars satisfying phase differences of $\pi/2$ and $\pi$, respectively.

polarization components ($|\mathbf{L}\rangle$ and $|\mathbf{R}\rangle$) carrying different OAM modes[1]:

$$|\psi\rangle = \cos\left(\frac{\theta}{2}\right)|m\rangle|\mathbf{R}\rangle + e^{i\alpha}\sin\left(\frac{\theta}{2}\right)|n\rangle|\mathbf{L}\rangle \tag{1}$$

where $\theta$ and $\alpha$ (shown in Fig. 2a) represent the weighted amplitude parameter and relative phase contributions of the circular polarization components, respectively. $m$ and $n$ denote the topological charges of the OAM modes in the right- and left-handed circular polarization components, respectively. Equation (1) can be rewritten as Jones vectors in the linear polarization basis (x-linear and y-linear polarizations):

$$|\psi\rangle = \frac{1}{\sqrt{2}}\begin{bmatrix} \cos\left(\frac{\theta}{2}\right)e^{im\zeta} + \sin\left(\frac{\theta}{2}\right)e^{in\zeta}e^{i\alpha} \\ -i\left(\cos\left(\frac{\theta}{2}\right)e^{im\zeta} - \sin\left(\frac{\theta}{2}\right)e^{in\zeta}e^{i\alpha}\right) \end{bmatrix} \tag{2}$$

where $\zeta$ is the azimuthal angle in the transverse cross-section plane of a structured light field. Equation 2 can be used to define different vector beams on the HOPS with spatially variant polarization distributions, with some representative examples shown in Fig. 2a (from I to V). Note that the two orthogonal modes formed in rectangular meta-atom cross-section are almost completely linearly polarized, justifying the use of the Jones-matrix formalism (see Supplementary Note 3 for further details).

We now demonstrate the design of 3D polymeric metasurfaces for implementing vector beams on the HOPS. 3D laser-nanoprinted nanopillar waveguides in a polymer matrix were employed as meta-atoms (Fig. 2b). To achieve the strong birefringence necessary for the

polarization control, we designed anisotropic nanopillars with rectangle cross-sections that support waveguide modes with distinctive indices for the polarizations along the short and long axes. According to the Jones matrix method, for an incident linear polarization along the x-direction (x-y-z coordinates are defined in Fig. 2b), the output light after passing through an anisotropic nanopillar waveguide is given as (Supplementary Note 4):

$$E_{out} = e^{i\varphi_x}\begin{bmatrix} t_x\cos^2(\gamma) + t_y\sin^2(\gamma)e^{i\Delta\varphi} \\ \frac{1}{2}(t_x - t_ye^{i\varphi})\sin(2\gamma) \end{bmatrix} \tag{3}$$

where $\gamma$ is the in-plane rotation angle of the nanopillar, $t_x$ and $t_y$ are the absolute transmission amplitudes of the transverse modes polarized in the x and y directions, respectively, and $\Delta\varphi$ is their relative phase. All these parameters are spatially dependent as functions of x and y. Equation 3 indicates that both the polarization (controlled by $\Delta\varphi$) and phase ($\varphi_x$) of an output beam can be controlled simultaneously by a single nanopillar, which forms the physical basis for implementing any arbitrarily structured light field. Note that $\varphi_x$ controls the transmitted phase from each nanopillar waveguide, while $\Delta\varphi$, because of the rectangular cross-section of the element, induces modal birefringence, together with the in-plane rotation angle $\gamma$ allow the control of the polarization of transmitted light. The key to our design is to find 3D nanopillars to be arranged in a particular distribution so that their optical responses can be precisely matched with any desired structured light field defined in Eq. 2.

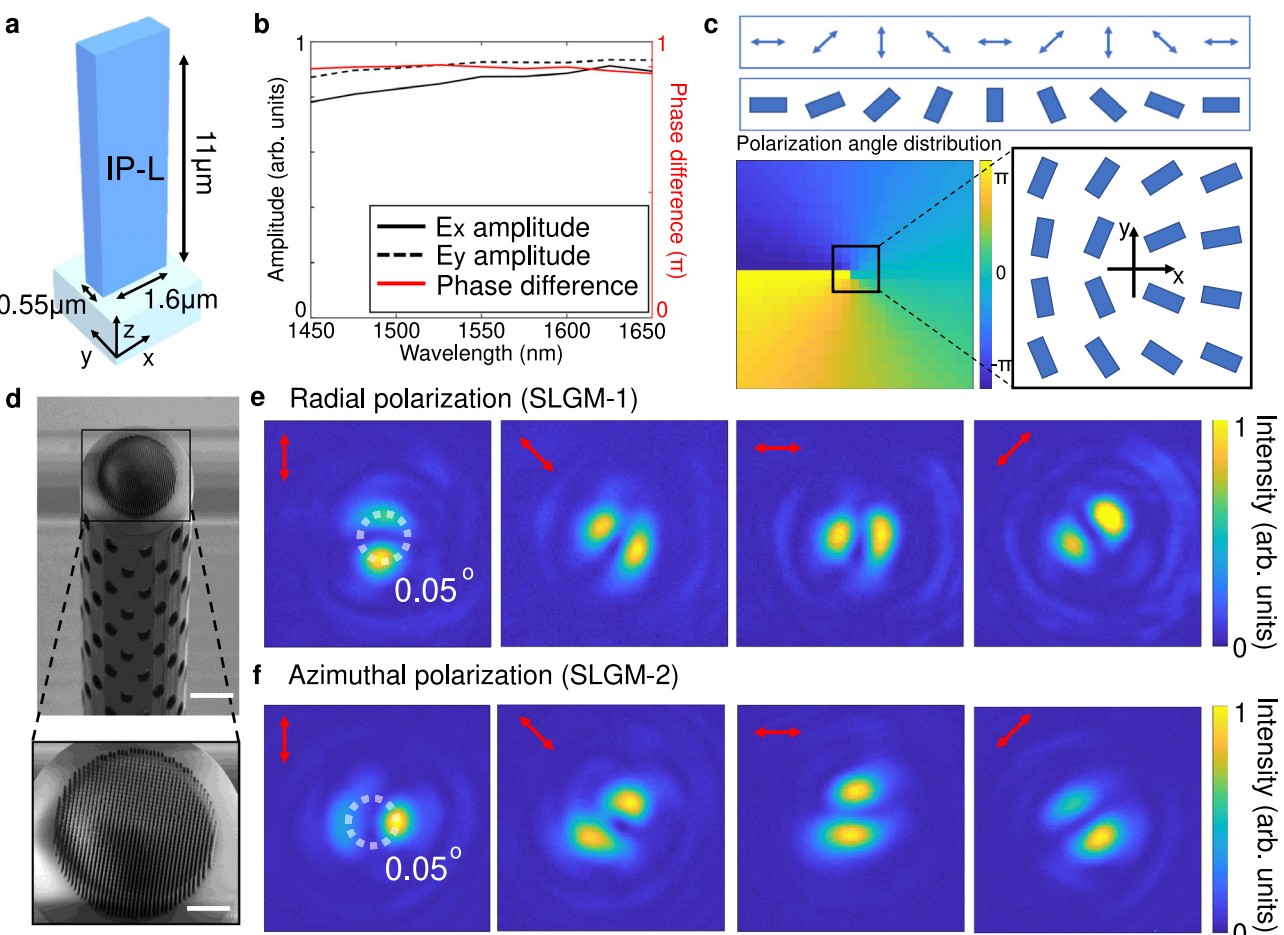

**Fig. 3 | Design and experimental characterization of SLGMs yielding cylindrical vector beam outputs. a** Schematic representation of the used 3D nanopillar meta-atom. **b** Simulated nanopillar response across a broad spectral range across the whole S, C and L telecommunication bands, as well as parts of the E and U bands. **c** Illustration of the in-plane angle distribution of 3D nanopillars used for creating the radial vector state. **d** Example SEM image of SLGM-1 used for creating the radial vector beam (scale bars: 50 μm (top) and 25 μm (bottom)). **e, f** Experimentally measured intensity distribution of the SLGM outputs of the radial (**e**) and azimuthal (**f**) vector beams in the Fourier plane (back focal plane imaging). The red arrows mark the polarization filtering axis of a linear polarizer inserted in front of the camera used for recording the polarization-dependent intensity profiles.

We have established a 3D meta-atom library based on the rigorous coupled-wave analysis method at a telecommunication wavelength of 1550 nm (Fig. 2c). It should be mentioned that the height degree of freedom of 3D nanopillars leads to a greatly expanded meta-atom library with a 3D dataset, which provides an expanded source to precisely match any desired polarization and phase responses. We present our simulation results of the phase difference between the x- and y-polarized transverse modes and the propagation phase of the x-linear polarization. We considered 3D nanopillars made of a lossless, commercial polymer material IP-L with an index of 1.5 (Nanoscribe, GmbH), having a fixed pitch distance of $P = 2.2$ μm, a length $L$ in the range of 0.8–1.8 μm, a width $W$ of 0.5–1.1 μm, and a height $H$ of 6–14 μm. Both the polarization eigenstates feature high transmission efficiency. 3D nanopillars exhibit a wide dynamic range from 0 to π in the phase difference and hence can function as wave-plates converting an incident x-linear polarization into different polarization outputs. Specifically, when the phase difference is equal to π, the corresponding nanopillars operate as half-wave plates that can rotate the angle of incident linear polarization, paving the way for generating various cylindrical vector beams on the equator of the HOPS. When the phase difference is π/2, in contrast, each nanopillar operates as quarter-wave plate and convert the linear polarization input into circular polarization, allowing access to the poles of the HOPS. When the phase difference takes any other value,

the nanopillars can convert the linear polarization into different elliptical polarizations and hence can reach to any desired state on the HOPS.

To better illustrate a wide coverage of 3D meta-atoms in phase difference and propagation phase responses, we highlight three individual planes in our 3D library dataset with different heights of $H = 8$, 10, and 12 μm in Fig. 2d. As an example, we can select 3D nanopillars of different heights to function like half- or quarter-wave plates, and in the meantime, they can cover the full range (0 to 2π) of the phase response. Therefore, our results suggest that 3D nanopillars with the unlocked height degree of freedom provide a powerful platform for implementing arbitrarily structured light. It is worth stressing that the ability to assign an individual height to each nanopillar is one of the key advantages of the nanoprinting process over other planar lithographic methods, which typically yield nanostructures of identical height[10,60] (Details in Supplementary Note 5). Lateral dimensions of the printed nanopillars were precisely characterized in Supplementary Note 6. Without loss of generality, we picked up five different vector states on the HOPS to prove our SLGM concept, including cylindrical vector beams on the equator of the first-order HOPS (Fig. 3), circularly polarized vortex beams carrying different OAM modes sitting on the poles of the first- and third-order HOPS (Fig. 4), and an arbitrary vector state on the HOPS carrying a spatially variant elliptical polarization distribution (Fig. 5).

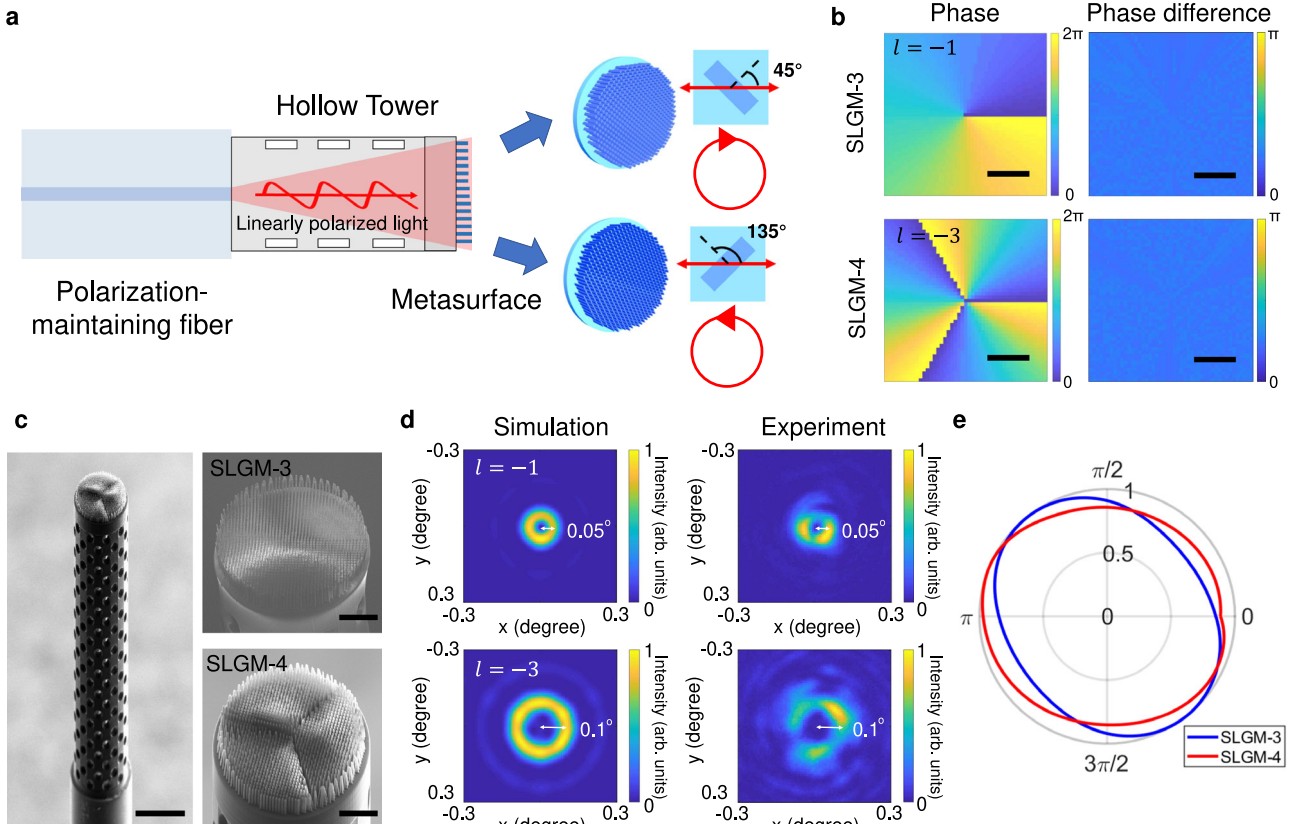

**Fig. 4 | Design and characterization of metafibers that generate circularly polarized vortex beams. a**, **b** Schematic illustration of creating circularly polarized vortex beam outputs on metafibers (a), which is achieved from arrays of 3D nanopillars with in-plane rotation angles of 45 and 135 degrees The specific propagation phase ($\varphi_x$) and phase difference ($\Delta\varphi$) maps for creating circularly polarized vortex beams of $|\psi_{R,-1}\rangle$ and $|\psi_{L,-3}\rangle$ are shown in (**b**). Scale bars in (**b**) represent 25 μm. **c** SEM images of the fabricated metafibers of SLGM−3. Scale bar (left panel): 100 μm. Right panel left and right: zoom-in areas of SLGM-3 and SLGM-4. Scale bars: 25 μm (**d**) Simulated (left column) and experimental (right column) results of the intensity distributions in the Fourier plane of the two SLGMs. The dashed circles mark the beam sizes in the Fourier plane. **e** Polarization ellipticity analysis of the SLGMs measured by inserting a rotating linear polarizer before the camera. The grey curve marks a perfectly circular polarized output.

## Cylindrical vector beams on SLGMs

Cylindrical vector beams, located at the equator of the first-order HOPS ($m = -n = 1$), are a special group of spatially variant vector beams with localized linear polarization. As typical examples of cylindrical vector beams, we experimentally demonstrate two SLGMs that can produce radial (SLGM-1) and azimuthal (SLGM-2) polarizations (Fig. 3). To obtain these states, the weighted amplitude parameter $\theta$ in Eq. 2 are set to $\frac{\pi}{2}$ and relative phase $\alpha$ to 0 and π, respectively. The resulting radial and azimuthal polarization states are given as $|\psi_r\rangle = \begin{bmatrix} \cos\zeta(x,y) \\ \sin\zeta(x,y) \end{bmatrix}$ and $|\psi_a\rangle = \begin{bmatrix} \sin\zeta(x,y) \\ -\cos\zeta(x,y) \end{bmatrix}$, respectively, where $\zeta(x,y)$ is the azimuthal angle in the transverse cross-section of the vector beams. To satisfy the only required polarization distributions, we used a single-sized 3D nanopillar (Fig. 3a) that behaves like a half-wave plate with a measured conversion efficiency of around 67%. (Supplementary Note 7) The nanopillar waveguide has a length $L$ of 1.60 μm, a width $W$ of 0.55 μm, and a height $H$ of 11 μm. We show that this nanopillar features both high transmission efficiency and π phase difference between the x- and y-linear transverse modes across the entire S, C and L telecommunication bands, from 1.45 to 1.65 μm (Fig. 3b). Equation 3 can then be simplified as $E_{out}(x,y) = \begin{bmatrix} \cos[2\gamma(x,y)] \\ \sin[2\gamma(x,y)] \end{bmatrix}$ by assuming $t_x \approx t_y \approx 1$. As such, for the radial polarization output, its localized linear polarization angle $\zeta(x,y)$ can be easily controlled by the in-plane rotation angle $\gamma(x,y)$ of the nanopillar (Fig. 3c). To achieve the azimuthal polarization output,

we can simply rotate all the nanopillars used for the above radial polarization by 45 degrees.

The cylindrical vector beam metasurface with a diameter of 100 μm was 3D laser nanoprinted on top of a hollow tower structure (height of 550 μm) that was interfaced on the PM-SMF end-face (Methods). To realize the correct polarization manipulation, the polarization axis of the PM-SMF must be carefully calibrated and aligned with the x-axis of the metasurface. This alignment was experimentally performed under a bright-field imaging microscope in the laser nanoprinting system (Nanoscribe GT). The metasurface is supported by a thin film (thickness 20 μm) printed together with nanostructure and tower. The side-view scanning electron microscope (SEM) image of SLGM-1 with the radial polarization output is given in Fig. 3d and Supplementary Note 8, revealing a well-defined 3D nanopillar metasurface on top of the tower. To characterize the cylindrical vector beams on SLGMs, a linearly polarized 1550 nm laser beam from a supercontinuum laser source (SuperK Fianium, NKT Photonics) and an infrared wavelength selector (SuperK Select, NKT Photonics) was coupled into the unstructured ends of SLGMs. The SLGM outputs were characterized using a home-built Fourier plane imaging setup (Supplementary Note 9) and recorded with a near-infrared camera (Raptor, Owl 640 M). Placing a linear polarizer in front of the camera results in two-lobe intensity patterns with respect to the axis of the linear polarizer, allowing us to identify the SLGM outputs as the radial (Fig. 3e) and azimuthal (Fig. 3f) vector beams[62]. Specifically, the lobes follow the orientation of the polarization axis of the linear

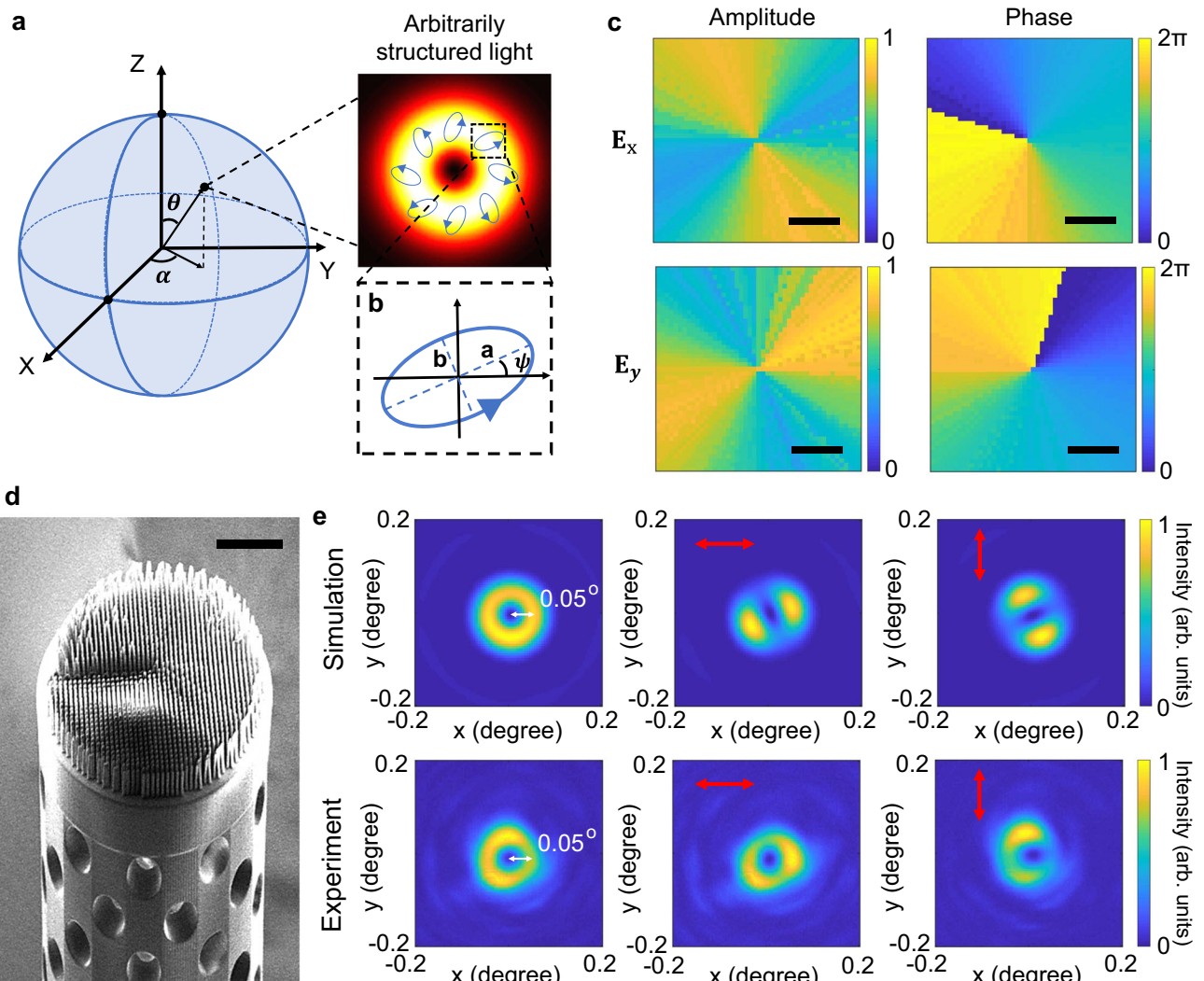

**Fig. 5 | Design and characterization of a metafiber (SLGM-5) used for creating an arbitrarily structured light field on the first-order HOPS. a** Schematic representation of the desired arbitrarily structured light field on the HOPS including the intensity and polarization distributions. **b** Close-up view of a local vector state of the structured light field, in which the ellipticity ratio and polarization direction angle are defined as $a/b$ and $\psi$, respectively. **c** The amplitude and phase distributions of both polarization components, based on optical responses of selectively matched 3D meta-atoms. Scale bars: 25 μm. **d** SEM image of SLGM-5. Scale bar: 30 μm. **e** Simulation and experimentally measured intensity distributions of SLGM-5 in the Fourier plane. Total intensity (first column) and polarization filtered intensity images (second and third columns, in which red arrows label the polarization axis of the linear polarizer). The dashed circles mark the beam divergence angle of 0.05.

polarizer, confirming the generation of cylindrically polarized beams. Note that due to the relatively large extension of the optical beam at the location of the metasurface, the local light intensity is sufficiently low to prevent damaging of the sample. Further details can be found in Fig. S17 of ref. 60.

**Circularly polarized vortex beams on SLGMs**

Circularly polarized vortex beams sitting at the poles of the HOPS represent the base vector states of the light fields generalized on the HOPS (see Eq. 1). We now demonstrate two additional SLGMs that enable the generation of right- and left-handed circularly polarized vortex beams with topological charges of −1 and −3, respectively (SLGM-3 and SLGM-4) (Fig. 4). For the circularly polarized vortex beam at the north pole of the first-order HOPS ($m = -n = 1$), both the weighted amplitude parameter $\theta$ and relative phase $\alpha$ in Eq. 2 are set to 0 (yielding sample SLGM-3). The corresponding circularly polarized vortex beam can be described mathematically as $|\psi_{R,-1}\rangle = \frac{1}{\sqrt{2}}e^{-i\zeta(x,y)}\begin{bmatrix}1\\-i\end{bmatrix}$. Similarly, for the circularly polarized vortex

beam at the south pole of the third-order HOPS ($m = -n = 3$, for SLGM-4), the weighted amplitude parameter $\theta$ and relative phase $\alpha$ in Eq. 2 are set to 0 and π, respectively, and the resulting beam is given as $|\psi_{L,-3}\rangle = \frac{1}{\sqrt{2}}e^{-i3\zeta(x,y)}\begin{bmatrix}1\\i\end{bmatrix}$. To realize such vector states through a metasurface, independent control of both polarization and propagation phase is necessary.

We designed nanopillars to function like quarter-wave plates, having high transmission efficiency ($t_x \approx t_y \approx 1$) as well as a phase difference of $\frac{\pi}{2}$ between the x- and y-linear polarization modes. According to Eq. 3, the nanopillar output should be expressed as $E_{\text{out}}(x,y) = e^{i\varphi_x(x,y)}\begin{bmatrix}\cos^2[\gamma(x,y)] + i^*\sin^2[\gamma(x,y)]\\\frac{1}{2}(1-i)\sin[2\gamma(x,y)]\end{bmatrix}$. By setting the in-plane rotation angle $\gamma(x,y)$ of the nanopillars to $\frac{\pi}{4}$ or $\frac{3\pi}{4}$, the outputs become right- or left-handed circularly polarized beams in the forms of $e^{i\varphi_x(x,y)}\begin{bmatrix}1\\-i\end{bmatrix}$ and $e^{i\varphi_x(x,y)}\begin{bmatrix}1\\i\end{bmatrix}$, respectively (Fig. 4a). Moreover, the

propagation phase of x-linear polarization $\varphi_x$ can satisfy a helical phase distribution $e^{il\zeta}$, where $l$ denotes the topological charge of the OAM mode. Owing to the greatly extended 3D meta-atom library, we can find 3D nanopillars with the propagation phase response from 0 to $2\pi$ to imprint different OAM modes while maintaining a nearly constant phase difference of $\frac{\pi}{2}$ (Fig. 4b).

We used the same fabrication process as described above to print two more SLGMs with circularly polarized vortex beam outputs. The side-view SEM images of these SLGMs are shown in Fig. 4c. To ensure precise polarization conversion, the x-axis of nanopillars were carefully aligned with respect to the polarization axis of the PM-SMFs by rotating them 45 and 135 degrees for the right- and left-handed circular polarization outputs, respectively. To characterize the helical beams of the fiber outputs, we recorded a series of intensity maps at different propagation distances from the metasurface plane up to 1 mm distance in real space (Supplementary Note 10). Due to the divergent wavefront of the fiber output leaving the PM-SMF, the OAM intensity distributions are enlarged as the propagation distance is increased. To further verify the OAM indices, we measured the Fourier plane image of the SLGMs. We found that doughnut-shaped beam patterns in the Fourier plane exhibit consistent divergence angles (SLGM3: 0.05° and SLGM4: 0.1°) with our simulation results, corroborating the nature of the transformed OAM beams (Fig. 4d). The inhomogeneity of the intensity distribution in the experimental results is mainly due to misalignment between the fiber output beam and metasurface (Supplementary Note 11). To consider only the OAM-induced beam divergence, we experimentally characterized the OAM beam divergence at a shifted Fourier plane that has the smallest beam sizes, through which we the fiber divergence is compensated. In addition, we noticed some interference effects in the intensity patterns, which we believe are mainly due to imperfect fiber cleaving such that the fiber output was not uniformly incident on the metasurfaces. Finally, we placed a linear polarizer in front of the camera to verify the circular polarization outputs. Our experimental results indicate the successful generation of circularly polarized vortex beam outputs in both SLGMs with high degree of circular polarizations (Fig. 4e).

### Arbitrarily structured light on SLGMs

To further demonstrate the superiority of 3D metasurfaces, we randomly chose an arbitrary state on the first-order HOPS ($m = -n = 1$) and demonstrate the transformation of such state by designing, fabricating, and employing another SLGM (SLGM-5) (Fig. 5a). The weighted amplitude parameter $\theta$ and relative phase $\alpha$ in Eq. 2 are set as $2\tan^{-1}\left(\frac{1}{3}\right)$ and $\frac{\pi}{4}$, respectively, leading to a vector state written as $|\psi\rangle = \frac{\sqrt{2}}{20}\begin{bmatrix} 3e^{i\zeta(x,y)} + e^{-i(\zeta(x,y)-\frac{\pi}{4})} \\ -i(3e^{i\zeta(x,y)} - e^{-i(\zeta(x,y)-\frac{\pi}{4})}) \end{bmatrix}$. It corresponds to an elliptically polarized state with an ellipticity ratio (a/b defined in Fig. 5b) of 2 and the polarization directions $\psi$ (semi-major axes) are spatially related to azimuthal angle $\zeta(x, y)$. The specific amplitude and phase distributions for the polarization states along x and y directions are displayed in Supplementary Note 12. Based on vectorial diffraction theory[63–65], we numerically simulated the diffraction pattern of the vector beam in a Fourier plane, with the polarization components along the x and y directions. To satisfy the desired amplitude and phase distributions, we selected 3D nanopillars with matched optical responses based on Eq. 3 to fulfill the desired amplitude and phase requirements (Fig. 5c). We also calculated the intensity response of our designed meta-atoms in the Fourier plane, which shows great agreement with our theoretical results. The resultant doughnut-shaped total intensity in the Fourier plane and the rotated two-lobe polarization patterns are consistent with our desired vector state (as shown in Fig. 5a). As such, the newly fabricated metafiber, SLGM-5 (Fig. 5d), was experimentally characterized by measuring the total intensity and polarization filtered images in the

Fourier plane of the metafiber (Fig. 5e), finding a good agreement with our theoretical and simulation results. Our simulation and experimental results show a consistent divergence angle of 0.05° induced by the first-order OAM beam of the SLGM-5 output. As such, we have experimentally verified that our metafiber platform is able to transform an arbitrary structured light field directly on the end face of a PM-SMF.

## Discussion

We have demonstrated a metafiber platform that can transform the output of a single-mode fiber into arbitrarily structured light on the HOPS using nanoprinted metasurfaces. We have successfully created polymeric 3D metasurfaces on commercial PM-SMFs. The unleashed height degree of freedom in 3D nanopillar meta-atoms offers a greatly expanded 3D meta-atom library, leading to independent, complete, and precise polarization and phase control at the level of individual meta-atoms. Several SLGMs were designed, 3D laser nanoprinted, and characterized, allowing for the on-fiber realization of five representative structured-light fields on the HOPS. These include radial and azimuthal polarizations (SLGM-1 and SLGM-2 in Fig. 3), circularly polarized vortex beams (SLGM-3 and SLGM-4 in Fig. 4), as well as an arbitrary vector state selected from the HOPS that carries a spatially variant elliptical polarization distribution (SLGM-5 in Fig. 5). Due to its simple and integrated nature, the implementation of structured light directly on optical fibers provides a paradigm for advancing optical fiber science and technology towards multimode light shaping and multi-dimensional light-matter interactions[66–73]. Structured light has found profound impact on both exploiting new phenomena in light-matter interactions[74] and offering a new multiplexing scheme for optical imaging and data storage[75]. Recently, helical beams carrying orbital angular momentum have shown efficient chiral sensitivity[18,76], which may open new opportunities in chiroptical spectroscopy. In a conventional structured light microscope, to prepare different structured light modes, a set of bulky and costly optical components, such as waveplates, lenses, and spatial light modulators must be cascaded on a table-based system. This significantly increases the system's complexity and cost. On the other hand, our structured light metafibers can directly generate arbitrary structured light fields on the hybrid-order Poincaré sphere, reducing the system complexity and cost. More importantly, owing to the use of single-mode fibers, our transformed arbitrary structured light modes can be easily delivered to a distant end without suffering from transmission or bending loss, mode distortion, and beam divergence. Therefore, we believe the on-demand generation of arbitrary structured light fields and the "plug and play" feature of our metafibers could add significant benefits to optical microscopes. For instance, long-distance transmission and delivery of structured light modes may not require sophisticated multimode fibers, which unfortunately suffer from intrinsic modal crosstalk and polarization mixing. Alternatively, our demonstrated metafiber platform could be used to realize arbitrarily structured light transformation at the fiber end-faces and achieve well-defined structured light modes of high quality without suffering susceptibility to bending or lack of reproducibility. Therefore, we believe that our demonstrated structured-light metafibers could find important applications, such as but not limited to fiber communications[15], fiber lasers[38], fiber sensors[39], endoscopic imaging[40], fiber lithography[41], and lab-on-fiber technology[42,43].

## Methods

### Fabrication of SLGMs

3D laser nanoprinting of polymer-based SLGMs was realized from a two-photon polymerization process via a commercial photolithography system (Photonic Professional GT, Nanoscribe GmbH).

Before nanoprinting, the fibers had to be prepared as follows: First, the coating of the PM-SMF was stripped and the end face was cleaved to ensure that the surface is complete and clean. The processed fiber was fixed in a fiber holder marked with lines to indicate the polarization direction of the fiber. The holder together with a fiber were put into the sample holder of the 3D laser nanoprinting system.

To precisely find the fiber and mark its exact 3D locations (x, y, and z coordinates), we found the fiber interface firstly with a low magnification (20x) air objective, followed by a high numerical aperture, oil immersion objective (Plan-Apochromat 63x/1.40 Oil DIC, Zeiss). Thereafter, the IP-L 780 photoresist resin was dropped cast onto the fiber and a high-precision translational stage was used for realignment. After the fiber end face has been found with the high numerical aperture objective, the whole structure that includes a 3D hollow tower (height of 550 μm) and a 3D metasurface was sequentially printed on top of the fiber in the dip-in configuration. We created some holes on the tower structure, ensuring that unpolymerized photoresist can be washed away during the chemical development process, and therefore creating a free-space beam expansion area inside the tower. The tower was designed to be hollow and to have a geometry such that the diffracted beam that is exiting the fiber remains unaffected (see Fig. 1(c)). In detail, the diameter of the circular hollow section of the tower is 90 μm (wall thickness 90 μm), while the beam coming out of the fiber has a diameter of 100 μm at the location of the metasurface. To increase the mechanical stability of the high-aspect-ratio polymer nanopillars, we have used the following optimized printing parameters: laser power 55 mW, scanning speed 3500 μm/s, and small hatching (lateral laser movement step: 10 nm) and slicing (axial laser movement step: 20 nm) distances. The nanoprinting process takes a total of 10–13 hours per sample, including 5 hours for the hollow tower and 5–8 hours for the metasurface depending on the design. After laser exposure, the samples were developed by immersing them in propylene glycol monomethyl ether acetate (PGMEA, Sigma-Aldrich) for 20 min, Isopropanol (IPA, Sigma-Aldrich) for 5 min, and Methoxy-nonafluorobutane (Novec 7100 Engineered Fluid, 3 M) for 2 min, respectively. The nanopillars reveal a mechanical long-term stability, since the implemented SLGMs show no signs of performance deterioration after 6 months of storage in air. This behavior agrees with our experience with strand-based light-guiding structures (so-called light cages[77,78]) that include polymer cylinders of even higher aspect ratios.

## Data availability
All data needed to evaluate the conclusions of the paper are available upon requests.

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

## Acknowledgements

This work was funded by the Deutsche Forschungsgemeinschaft (DFG, German Research Foundation) under grant numbers MA 4699/7-1, MA

4699/9–1, SCHM2655/15-1, SCHM2655/21-1, SCHM2655/23-1, and the Center for NanoScience (CeNS). H. R. acknowledges support from the Australian Research Council DECRA Fellowship (project DE220101085). H.R. and S.A.M. acknowledge support from the Australian Research Council Discovery Project (project DP220102152). C. L. acknowledges the scholarship support from the China Scholarship Council. A. M. acknowledges support from projects PROMETEO/2021/006 (Generalitat Valenciana, Spain) and PID2021-123124OB-I00 (Ministerio de Ciencia e Innovación, Spain). S.A.M. additionally acknowledges the Lee-Lucas Chair in Physics. This work was performed in part at the Melbourne Centre for Nanofabrication (MCN) in the Victorian Node of the Australian National Fabrication Facility (ANFF).

## Author contributions

C.L. and H.R. conceived the idea; C.L. performed the numerical analysis, fabrication, and experimental characterization; T.W. contributed to fiber characterization; F.J.W., A.M., and L.d.S.M. contributed to optical measurement setup; C.L., S.A. Maier, M.A.S., and H.R. contributed to data analysis; H.R. provided simulation source files; C.L. and H.R. wrote the paper with contributions from all authors.

## Competing interests

The authors declare no competing interests.
