## [Peer Review File · Nature Communications]

Metafiber transforming arbitrarily structured lightREVIEWER COMMENTS

Reviewer #1 (Remarks to the Author):

In the manuscript "Metafiber transforming arbitrarily structured light", the authors have demonstrated a novel metafiber platform capable of transforming the output of a single-mode optical fiber into arbitrarily structured light on the HOPS using nanoprinted metasurfaces. The platform achieves precise polarization and phase control through 3D nanopillar meta-atoms and successfully generates various structured light fields, including radial and azimuthal polarizations, circularly polarized vortex beams, and spatially variant elliptical polarization distributions. The platform is simple and integrated, providing a new paradigm for advancing optical fiber science and technology, with potential applications in fiber communications, fiber lasers, fiber sensors, endoscopic imaging, fiber lithography, and lab-on-fiber technology. The manuscript is well organized, and the simulations are in good agreement with the measurements. I think that the manuscript needs revisions and clarifications before publication.

1. A hollow tower structure (with diameter of 100 μm , height of 550 μm) was first laser printed onto the fiber end-face. What is the material used for the 3D hollow tower in the fabrication process of SLGMs? What is the distribution of the light field at the exit face of the fiber before the 3D hollow tower? How does the light field distribution change after the addition of the 3D hollow tower?
2. In Figure 3(e) and (f), Are the light field distributions the near fields? If there are far fields, what are the distances between the testing positions and the metasurface? Could you please add a colorbar to Figure 3(e) and (f) to provide additional information about the intensity values in the images?
3. The SEM image of SLGM-1 in Fig3(d) is not clearly enough, I would like to suggest increasing the dimension of the images to allow for better observation of the morphology of the nanopillars.
4. In Fig2, the authors have presented three exemplary data planes in the meta-atom library with heights of $H=8, 10, 12 \mu\text{m}$. I would like to inquire about the fabrication process used to control the height of the rectangular nanopillars on the metasurface, as it is evident that the height of the nanopillars affects the phase response of the X and Y polarizations. Additionally, how is the height parameter of the fabricated nanopillars measured and characterized?
5. In the description of the fabrication process of SLGMs using 3D laser nanoprinting, it is mentioned that the optimized printing parameters, including laser power and scanning speed, were used. However, the manuscript does not provide a detailed explanation of the basis for the selection of these parameters and their impact on the performance of SLGMs. Could the authors further explain the criteria for the selection of these parameters and how they affect the performance of SLGMs (e.g., transmission efficiency, polarization control precision, etc.)?
6. The manuscript mentioned the use of a 3D hollow tower structure and the creation of holes in the tower structure to ensure that unpolymerized photoresist can be washed away. Could the authors elaborate about the design principle of this 3D hollow tower structure and its role in the overall fabrication process of SLGMs? Additionally, does this structure have any impact on the quality of the light beam output from the fiber, such as beam divergence, beam distortion, etc.?

Reviewer #2 (Remarks to the Author):

This paper reports the integration of metasurfaces onto the endface of polarization-maintaining single-mode fibers to transform a linear polarized light wave into different structured light fields. The work include the design and fabrication by 3D laser printing of nanostructures on the fiber endface and characterization of the integrated metafibers. The work is very interesting and could be a useful step forward for applications of structured light in fiber optic systems.

I have the following comments for the improvement of the paper.

A hollow tower structured was printed onto the fiber endface to expand the beam to ~100 micrometers, followed by a thin supporting film (shown in Fig.1b but not described in the paper), and then the not meta atoms. What is the thickness of the thin film and would it affect of generation of the structured light field? Can you comment on the advantage of using this hollow tower over a graded index lens popularly used for light collimation from a single mode fiber?

The meta atoms used are rectangular waveguides with width and length on the order of 0.5 to 1.8 micrometers. For these sub-micrometer waveguides, the eigen modes are not perfectly linear and could have significant cross-polarization component. Therefore the representation of the structured light field with Jones vectors on the linear polarization basis would have errors. Could the authors comment on accuracy of approximating the light field with by the linear Jones vectors?

The experimentally measured intensity patterns of the SLGM outputs of the radial (Fig.3e) and azimuthal (Fig.3f) vectors deviate from the expected symmetric two-lobe patterns. What is the reason behind this, it is partly due to the approximation of the field vectors of the meta atoms by linear vectors? Or it is due to imperfection of the fabrication of the metal atoms? Some comments would help to reader to understand. I suggest to include the simulated results, compare with the measured ones and discuss the differences.

The simulated and the measured intensity patterns in Fig.4 and 5 are also quite different.

It is would be nice to state at least one specific application of the device.

In Supplementary Note 1, with the authors' definition of matrix M , $E_{out} = RMR^{-1} E_{in}$ seems incorrect. R and R^{-1} are already included in M ?

Reviewer #3 (Remarks to the Author):

The manuscript titled "Metafiber: Transforming Arbitrarily Structured Light" presents a novel metafiber platform capable of generating arbitrarily structured light output on the HOPS through the use of nanoprinted metasurfaces. The researchers employed polymeric metasurfaces and laser nanoprinting techniques to achieve this breakthrough and successfully interfaced it with single-mode fibers. This platform represents a significant advancement over traditional optical fiber technology and holds enormous potential for various applications. While the results are impressive, there remain several important and unresolved issues that require further exploration.

Here are the detailed comments:

1. There have been related articles to implement OAM transmission on optical fiber, such as Laser & Photonics Reviews 15.5 (2021): 2000581. Optica 9.6 (2022): 645-651. Appl. Sci. 2019, 9(5), 1033, so the expression of "To date, the generation of structured light is still handled outside the fiber via bulky optics in free space" should be modified more exactly.
2. The aspect ratio of the nanopillars is very high, so, as we all know, the antenna may collapse very easily. Using 3D printing technology, how accurate is the structure, and how about the morphology and stability of the structure?
3. In the paper, the height of 3D nanopillars are unlocked in one metasurface. The phase response of the nanopillar is related to the height. Whether crosstalk exists in the phase response between adjacent nanorods of different heights in the same structure? What are the differences and advantages between the same height nanopillars and the different height nanopillars in one metasurface?
4. Time spent is an important indicator of 3D printing. How long does it take to print the whole structure?
5. Why Eq(3) indicates that both the polarization and phase of an output beam can be controlled simultaneously by a single nanopillar, please give further detailed derivation.

6. How much power the structure can withstand and how efficient the structure is to light conversion, please give further supplements.

Reviewer #1 (Remarks to the Author)

In the manuscript “Metafiber transforming arbitrarily structured light”, the authors have demonstrated a novel metafiber platform capable of transforming the output of a single-mode optical fiber into arbitrarily structured light on the HOPS using nanoprinted metasurfaces. The platform achieves precise polarization and phase control through 3D nanopillar meta-atoms and successfully generates various structured light fields, including radial and azimuthal polarizations, circularly polarized vortex beams, and spatially variant elliptical polarization distributions. The platform is simple and integrated, providing a new paradigm for advancing optical fiber science and technology, with potential applications in fiber communications, fiber lasers, fiber sensors, endoscopic imaging, fiber lithography, and lab-on-fiber technology. The manuscript is well organized, and the simulations are in good agreement with the measurements. I think that the manuscript needs revisions and clarifications before publication.

We thank the Reviewer for the positive evaluation of our manuscript and on the findings presented. In the following, we address all comments in detail and mention the relevant changes in the main text and in the Supplementary Information.

R1.1: A hollow tower structure (with diameter of 100 μm , height of 550 μm) was first laser printed onto the fiber end-face. What is the material used for the 3D hollow tower in the fabrication process of SLGMs? What is the distribution of the light field at the exit face of the fiber before the 3D hollow tower? How does the light field distribution change after the addition of the 3D hollow tower?

We thank the Reviewer for raising the question about the influence of the 3D tower on the properties of the metafiber. Below, we address the various questions posed by the Reviewer separately.

Material: Kindly note that the tower and the metasurface were printed together in one run, i.e., they thus are made of the same material (here the polymer IP-L is used, which is provided by Nanoscribe). To emphasize this point in the manuscript, we have modified the main text as follows:

... of the PM-SMF to be freely expanded to fully cover the area of a 3D metasurface with a diameter of around 100 μm , a hollow tower structure (height of 550 μm) was first laser printed onto the fiber end-face (Fig. 1a). **Note that the tower and the metasurface were printed together in one run, i.e., they thus are made of the same material (here IP-L polymer).** To prove our concept ...

Distribution of light field at the fiber exit before the tower:

Based on the Reviewer’s comment regarding the distribution of the light field at the end face of the fiber before the tower, we characterized the output beam of the blank fiber (PM-SMF, PM1550-XP, Thorlabs). In detail, we measured the beam diameter ($\lambda_0 = 1550\text{nm}$) for different distances from the fiber surface and compared it with the theory of Gaussian beams (Fig. R1), allowing us to determine the size of the beam at the fiber surface [Bahaa E. A. Saleh, Malvin Carl Teich, Fundamentals of Photonics, 1991].

Fig. R1. Characterization of the output beam of the fiber used (PM-SMF, PM1550-XP, Thorlabs) without the tower. The points refer to the measured mode field diameters at selected distances from the fiber end face, while the line is a fit to the data points. The top images refer to the measured beam intensity profiles, with the respective numbers in the top left corners referring to the distance to the fiber surface (in μm). The red square in the graph indicates the height of the tower used in the metasurface-based experiments discussed in the manuscript.

Within the context of Gaussian beams, the beam radius is defined with respect to the transverse position at which the intensity has dropped to $1/e^2$ of the value at the center of the beam, being given by

$$w(z) = w_0 \sqrt{1 + (z/z_R)^2}$$

with the beam waist w_0 , the longitudinal position z and the Rayleigh length z_R . The latter is defined by $z_R = \pi w_0^2 / \lambda_0$ (vacuum wavelength λ_0). By fitting the measured data (points in Fig. R1) with the equation stated above, a beam diameter at the fiber surface of $w_0^{exp} = 10.1 \mu\text{m}$ is obtained, which fits well with the value from the datasheet $w_0^{data} = 10.1 \mu\text{m}$. It is important to mention that the measurements show field patterns with circular distributions at any distance from the fiber end face (top images in Fig. R1), which is important to consider in case metasurface structures are created on the tower.

To account for the Reviewer's comment, this figure together with corresponding discussion have been added to the Supplementary Information. We have also modified the main text as follows: ... we experimentally used a commercial polarization-maintaining single-mode fiber (PM-SMF, PM1550-XP, Thorlabs). **Details on the fiber used can be found in Supplementary Note 1.** To allow the output ...

Supplementary Note 1. Characterization of the beam at the fiber end face.

The characterization of the distribution of the light field at the end face of the fiber before the tower relies on characterizing the output beam of the blank fiber (PM-SMF,

PM1550-XP, Thorlabs). In detail, we measured the beam diameter ($\lambda_0 = 1550nm$) for different distances from the fiber surface and compared it with the theory of Gaussian beams (Fig. S1), allowing us to determine the size of the beam at the fiber surface¹.

Fig. S1. Characterization of the output beam of the fiber used (PM-SMF, PM1550-XP, Thorlabs) without the tower. The points refer to the measured mode field diameters at selected distances from the fiber end face, while the line is a fit to the data points. The top images refer to the measured beam patterns, with the respective numbers in the top left corners referring to the distance to the fiber surface (in μm). The red square indicates the height of the tower used in the metasurface-based experiments discussed in the manuscript.

Within the context of Gaussian beams, the beam radius is defined with respect to the transverse position at which the intensity has dropped to $1/e^2$, is given by

$$w(z) = w_0 \sqrt{1 + (z/z_R)^2},$$

with the beam waist w_0 , the longitudinal position z and the Rayleigh length z_R . The latter is defined by $z_R = \pi w_0^2 / \lambda_0$ (vacuum wavelength λ_0). By fitting the measured data (points in Fig. S1) with the equation stated above, a beam diameter at the fiber surface of $w_0^{exp} = 10.1 \mu\text{m}$ is obtained, which fits well with the value from the datasheet $w_0^{data} = 10.1 \mu\text{m}$. It is important to mention that the measurements show field patterns with circular distributions at any distance from the fiber end face (top images in Fig. S1), which is important to consider in case metastructures are created on the tower.

[1] B. E. A. Saleh, M. C. Teich, *Fundamentals of Photonics* (1991).

Modification of light field after the addition of tower: The tower was designed to be hollow to reduce its impact on the output light. To verify the influence of the hollow tower on the light field distribution, we measured the optical field distribution of single fiber (Fig. R1) and tower fiber at $550 \mu\text{m}$ (Fig. S3). We found that the inner diameter of the tower ($90 \mu\text{m}$) is smaller than the beam at the top part of the tower. As a result, some part of the beam is reflected and causes a ring-shaped interference pattern on the top of the tower, which might influence the

wavefront shaping process. Therefore, in this revision we simulated the influence of the interference pattern created by a tower fiber on the imaging results for both the polarization and phase singularity beams (Supplementary Note 2). We find that the ring-shaped interference is related to high spatial-frequency components in the Fourier plane and does not change the central part of the polarization and phase singularity beams in our imaging results, although accompanied with some reduced (by ~20%) amplitude modulation (Fig. S2).

We can further optimize our design of the hollow tower by including an extended upper region for larger distances from the fiber end face (Fig. S3). This allows for the inner hollow part of the tower to gradually increase up to a diameter of $130\mu\text{m}$ at the top, preserving the Gaussian mode profile of the fiber output. However, the new tower results are preliminary and require much more work to enhance its mechanical robustness for hosting metasurfaces, and therefore, we would opt not to use them in current work.

To account for the Reviewer's comment and clarify the influence of tower, we have added a Supplementary Note 2:

Supplementary Note 2. The effect of the interference pattern created by a tower fiber on the imaging results.

The tower was designed to be hollow to reduce its impact on the output light, although we found that the inner diameter of the tower ($90\mu\text{m}$) is smaller than the beam at the top part of the tower. As a result, some part of the beam is reflected and causes a ring-shaped interference pattern on the top of the tower, which might influence the wavefront shaping process (Fig. S2b). We simulated the interference of such a ring-shaped interference pattern on the imaging results of both the polarization and phase singularity beams in the Fourier plane. Our simulation shows that the ring-shaped interference is related to high spatial-frequency components in the Fourier plane and does not change the central part of the polarization and phase singularity beams in our imaging results (Figs. S2d and S2f).

Fig. S2. Simulation of influence of the ring-shaped interference pattern induced by a tower on the imaging results of both the polarization and phase singularity beams. (a and b) Intensity distributions of a Gaussian mode and a ring-shaped interference mode (due to a fiber tower reflection) at the metasurface plane, respectively (scale bar: 25 μm). (c and d) Intensity distributions of a polarization singularity beam (azimuthally polarized beam) in the Fourier plane under the illumination of a Gaussian mode (c) and a ring-shaped interference mode (d), respectively (e and f). The counterparts of (c and d) but with an OAM-carrying phase singularity beam of $l=3$. The dashed circles mark the beam divergence angle of 0.06 (c and d) and 0.1 (e and f) respectively.

In our experiment, we can further optimize our design of the hollow tower by including an extended upper region for larger distances from the fiber end face (Fig. S3). This allows for the inner hollow part of the tower to gradually increase up to a diameter of 130 μm at the top, preserving the Gaussian mode profile of the fiber output. However, the new tower results are preliminary and require much more work to enhance its mechanical robustness for hosting metasurfaces, and therefore, we would opt not to use them in current work.

Fig. S3. Intensity distributions of fiber outputs on the surface of hollow tower structures printed on the end faces of SMF-28 fibers. (a) The output intensity distribution of a straight hollow tower with a diameter of 90 μm . (b) The output intensity distribution of an expanded hollow tower that leads to an extended upper part with a diameter of 130 μm . Scale bar: 20 μm (c and d) Schematics and microscope images of the straight tower (c) and the enlarged tower (d), respectively. Scale bar: 50 μm .

R1.2: In Figure 3(e) and (f), Are the light field distributions the near fields? If there are far fields, what are the distances between the testing positions and the metasurface? Could you please add a colorbar to Figure 3(e) and (f) to provide additional information about the intensity values in the images?

We thank the Reviewer for the comments on Figure 3. Here the Reviewer is correct that the images show the near field distributions of light after the metasurface. Through appropriate imaging, we are collecting the light in an image plane just on the surface of the metasurface, leading to our previous imaging results shown in Figure 3. However, to maintain consistency within the other fiber results shown in Figures 4 and 5, we repeated the measurements related to SLGM-1 and SLGM-2 and decided to replace the original near-field images with the new experimental results related to back focal plane imaging (please see revised Figure 3, which corresponds to far-field imaging results). We also added the color bar to the new Figs.3(e) and (f).

To account for the Reviewer's comment, we have modified the main text as follows:

...The SLGM outputs were characterized using a home-built **Fourier plane imaging setup (Supplementary Note 9)** and recorded with a near-infrared camera (Raptor, Owl 640 M). Placing a linear polarizer in front of the camera results in two-lobe intensity patterns with respect to the axis of the linear polarizer, allowing us to identify the SLGM outputs as the radial (Fig. 3e) and azimuthal (Fig. 3f) vector beams. ...

Fig 3. Design and experimental characterization of SLGMs yielding cylindrical vector

beam outputs. (a) Schematic representation of the used 3D nanopillar meta-atom. (b) Simulated nanopillar response over a broad spectral range across the entire S, C and L telecommunication bands, as well as parts of the E and U bands. (c) Illustration of the in-plane angle distribution of 3D nanopillars used for creating the radial vector state. (d) Example SEM image of SLGM-1 used for creating the radial vector beam (scale bar: 25 μm). (e and f) Experimentally measured intensity distribution of the SLGM outputs of the radial (e) and azimuthal (f) vector beams in the Fourier plane (back focal plane imaging). The red arrows mark the polarization filtering axis of a linear polarizer inserted in front of the camera used for recording the polarization-dependent intensity profiles.

R1.3: The SEM image of SLGM-1 in Fig3(d) is not clearly enough, I would like to suggest increasing the dimension of the images to allow for better observation of the morphology of the nanopillars.

We thank the Reviewer for this comment. To reveal the morphology of the metasurface and the tower more clearly, we (i) replaced the SEM image in Fig. 3d by one with higher resolution (revised Fig. 3d), and (ii) provided more SEM images in Supplementary Note 8.

Supplementary Note 8. Zoom in image of printed metasurface.

To reveal the morphology of the metasurface and the tower more clearly, we have provided more SEM images of SLGM-1 in Fig. 3d (Fig. S9).

Fig. S9. Example SEM images of SLGM-1 used for creating the radial vector beam. (a) SEM image of the whole metasurface (scale bar: 20 μm). (b) Zoom-in area of (a) (scale bar: 8 μm).

R1.4: In Fig2, the authors have presented three exemplary data planes in the meta-atom library with heights of $H=8, 10, 12 \mu\text{m}$. I would like to inquire about the fabrication process used to control the height of the rectangular nanopillars on the metasurface, as it is evident that the height of the nanopillars affects the phase response of the X and Y polarizations. Additionally, how is the height parameter of the fabricated nanopillars measured and characterized?

We thank the Reviewer for raising the question about the nanopillars' height determination. Since our implementation relies on 3D nanoprinting, we can directly design the height of each nanopillar individually and print the array of nanopillars together with the tower in one go. Note

that the ability to assign an individual height to each nanopillar is one of the key advantages of the nanoprinting process over other planar lithography-based methods, which typically yield nanostructures of identical height. In the practical printing process, we used a small slicing distance of 20nm along the height dimension to precisely control the height of each individual nanopillar (Fig. S5).

To account for the Reviewer's comment, we have modified the text as follows: ... Therefore, our results suggest that 3D nanopillars with unlocked height degree of freedom provide a powerful platform for implementing arbitrarily structured light. **It is worth stressing that the ability to assign an individual height to each nanopillar is one of the key advantages of the nanoprinting process over other planar lithography methods, which typically yield nanostructures of identical height.^{10,60} (Details in Supplementary Note 5) ...**

[10] H. Ren, X. Fang, J. Jang, J. Burger, J. Rho, S. A. Maier, Complex-amplitude metasurface-based orbital angular momentum holography in momentum space, *Nat. Nanotechnol.* 15, 948-955 (2020).

[60] H. Ren et al., An achromatic metafiber for focusing and imaging across the entire telecommunication range, *Nat. Commun.* 13, 4183 (2022).

To clarify this important point, we have added some discussion in Supplementary Note 5:

Supplementary Note 5: Height calibration of nanopillars.

To calibrate the height of nanopillars, we fabricated nanopillars with height variation from 6 μ m to 10 μ m with an increment of 1 μ m, which cover most heights defined in our 3D design library. To make the measurement more accurate, we decided to push down the nanopillars instead of measuring height at a tilt angle of SEM imaging. It should be mentioned that the height resolution (smallest variation in height) is controlled by the slicing accuracy in the nanoprinting process, which was set to be 20nm in our experiment. We used this small slicing distance (much smaller than its typical use) to enhance the mechanical strength of high-aspect-ratio nanopillars to avoid collapsing. To provide a precise characterization, the actual procedure includes fabrication of nanopillars of different heights on a glass substrate, turning the sample upside down and placing it on a flat surface, and slightly sliding the sample. This forces the nanopillars to be pushed down on the surface. Undamaged pillars were measured under SEM and compared to the designed heights. Thus, we obtain a relationship between design and the actual height of the nanopillars. Selected SEM images of nanopillars and their corresponding heights are shown in Fig S5.

Fig. S5. Height calibration of nanopillars. Top images: SEM images of three selected nanopillars with designed heights of 6μm, 8μm and 10μm, respectively (scale bar: 3μm). The black solid line is the fitted relationship between designed height and the measured height.

The relationship between designed height and measured height is linearly fitted with the following expression, with an offset arising from the axial extension of the focal spot:

$$H_{measured} = H_{design} + 0.48\mu m.$$

R1.5: In the description of the fabrication process of SLGMs using 3D laser nanoprinting, it is mentioned that the optimized printing parameters, including laser power and scanning speed, were used. However, the manuscript does not provide a detailed explanation of the basis for the selection of these parameters and their impact on the performance of SLGMs. Could the authors further explain the criteria for the selection of these parameters and how they affect the performance of SLGMs (e.g., transmission efficiency, polarization control precision, etc.)?

We thank the Reviewer for the comments on the printing parameters. The printing recipe contains three parameters: laser power (exposure dose), scanning speed (exposure time) and scanning resolution (slicing and hatching distances), all of which are related to the degree of polymerization of the photoresist used during printing. A higher level of polymerization provides higher mechanical strength, resulting in high aspect ratios between 16 and 24 of our nanopillars. However, excessive laser energy can cause overexposure of the photoresist, while slow scanning speed may impose the entire printing process to be time-consuming. To take all these issues into account, we optimized the recipe for nanoprinting to find a balance between the realization of mechanically stable and complete structures and a reasonable printing time.

To account for the Reviewer's comment, we have modified the main text as follows:

...out of the fiber has a much smaller diameter at the location of the metasurface (100μm). **To**

increase the mechanical stability of the high-aspect ratio polymer nanopillars, we have used the following optimized printing parameters: laser power 55mW, scanning speed 3500 μ m/s, and small hatching (lateral laser movement step: 10nm) and slicing (axial laser movement step: 20nm) distances. The printing process takes a total of 10-13 hours per sample, including 5 hours for the 3D hollow...

R1.6: The manuscript mentioned the use of a 3D hollow tower structure and the creation of holes in the tower structure to ensure that unpolymerized photoresist can be washed away. Could the authors elaborate about the design principle of this 3D hollow tower structure and its role in the overall fabrication process of SLGMs? Additionally, does this structure have any impact on the quality of the light beam output from the fiber, such as beam divergence, beam distortion, etc.?

As described in our response to the Reviewer's first comment (R1.1), the purpose of the tower is to allow the beam leaving the fiber to sufficiently expand so that as many elements of the metasurface as possible contribute to the beam shaping. In detail, the diameter of the circular hollow section of the tower is 110 μ m (wall thickness 10 μ m), while the beam coming out of the fiber has a diameter of 100 μ m at the location of the metasurface. **For further details of the impact on the light beam**, we kindly refer the Reviewer's attention to our response to the first comment of the Reviewer (R1.1).

Reviewer #2 (Remarks to the Author)

This paper reports the integration of metasurfaces onto the endface of polarization-maintaining single-mode fibers to transform a linear polarized light wave into different structured light fields. The work include the design and fabrication by 3D laser printing of nanostructures on the fiber endface and characterization of the integrated metafibers. The work is very interesting and could be a useful step forward for applications of structured light in fiber optic systems.

We wish to convey our appreciation to the Reviewer for the examination of our manuscript. The comments offered have been key to refining and improving our work. In the subsequent section, we offer point-by-point responses, addressing each comment with care, and discuss the adaptations and changes we have incorporated to align with the feedback provided by the Reviewer.

I have the following comments for the improvement of the paper.

R2.1: A hollow tower structured was printed onto the fiber endface to expand the beam to ~100 micrometers, followed by a thin supporting film (shown in Fig.1b but not described in the paper), and then the not meta atoms. What is the thickness of the thin film and would it affect of generation of the structured light field? Can you comment on the advantage of using this hollow tower over a graded index lens popularly used for light collimation from a single mode fiber?

We thank the Reviewer for the important comments related to the tower structure, which we individually addressed in the following.

Thickness of film and impact on optical properties of metasurface: The thin film supporting the metasurface has a thickness of 20um thick and is printed together with nanostructure and tower. To account for the Reviewer's comment, we have modified the main text as follows:

... This alignment was experimentally performed under a bright-field imaging microscope in the laser nanoprinting system (Nanoscribe GT). **The metasurface is supported by a thin film (thickness 20um) printed together with nanostructure and tower.** The side-view scanning electron microscope (SEM) image of SLGM-1 with ...

To verify the effect of the hollow tower on the output beam, we measured light intensity distribution produced by a bare fiber and tower fiber. The experimental results (Fig. S3) show some reflections produced by the inner walls of the tower could cause ring patterns. (More details can be found in our response to the first comment of Reviewer #1 (R1.1)). However, simulations show that this ring pattern has no effect on the imaging results in the Fourier plane and it will not change the conclusion of the manuscript. To account for the Reviewer's comment and clarify the influence of tower, we have added a Supplementary Note 2:

Supplementary Note 2. The effect of the interference pattern created by a tower fiber on the imaging results.

The tower was designed to be hollow to reduce its impact on the output light, although we found that the inner diameter of the tower (90µm) is smaller than the beam at the

top part of the tower. As a result, some part of the beam is reflected and causes a ring-shaped interference pattern on the top of the tower, which might influence the wavefront shaping process (Fig. S2b). We simulated the interference of such a ring-shaped interference pattern on the imaging results of both the polarization and phase singularity beams in the Fourier plane. Our simulation shows that the ring-shaped interference is related to high spatial-frequency components in the Fourier plane and does not change the central part of the polarization and phase singularity beams in our imaging results (Figs. S2d and S2f).

Fig. S2. Simulation of influence of the ring-shaped interference pattern induced by a tower on the imaging results of both the polarization and phase singularity beams. (a and b) Intensity distributions of a Gaussian mode and a ring-shaped interference mode (due to a fiber tower reflection) at the metasurface plane, respectively (scale bar: $25\mu\text{m}$). (c and d) Intensity distributions of a polarization singularity beam (azimuthally polarized beam) in the Fourier plane under the illumination of a Gaussian mode (c) and a ring-shaped interference mode (d), respectively (e and f). The counterparts of (c and d) but with an OAM-carrying phase singularity beam of $l=3$. The dashed circles mark the beam divergence angle of 0.06 (c and d) and 0.1 (e and f) respectively.

In our experiment, we can further optimize our design of the hollow tower by including an extended upper region for larger distances from the fiber end face (Fig. S3). This allows for the inner hollow part of the tower to gradually increase up to a diameter of $130\mu\text{m}$ at the top, preserving the Gaussian mode profile of the fiber output. However, the new tower results are preliminary and require much more work to enhance its mechanical robustness for hosting metasurfaces, and therefore, we would opt not to use them in current work.

Fig. S3. Intensity distributions of fiber outputs on the surface of hollow tower structures printed on the end faces of SMF-28 fibers. (a) The output intensity distribution of a straight hollow tower with a diameter of $90\mu\text{m}$. (b) The output intensity distribution of an expanded hollow tower that leads to an extended upper part with a diameter of $130\mu\text{m}$. Scale bar: $20\mu\text{m}$ (c and d) Schematics and microscope images of the straight tower (c) and the enlarged tower (d), respectively. Scale bar: $50\mu\text{m}$.

Hollow tower vs. graded index lens: Generally, the utilization of a graded index lens is a widely used and reliable approach for the collimating optical beams. However, in this work, we propose a design concept that prioritizes simplicity and minimal complexity through the employment of free diffraction for the beam exiting the fiber. In our design, we eliminate the need for additional optical components on the surface of the fiber by taking advantage of free diffraction. Note that in this work, we did not include the phase profile of a lens in our metasurface design to collimate the beam, since our priority is to demonstrate the generation of complex beams using a fiber-integrated system. We should mention that the phase profile of a lens can be implemented onto our metasurface design when a collimated output is demanded, which is empowered by the simultaneous and independent control of both the polarization and phase responses at a single meta-atom (nanopillar) level.

To account for the Reviewer's comment, we have modified the main text as follows:

... To allow the output of the PM-SMF to be freely expanded to fully cover the area of a 3D metasurface with a diameter of around $100\mu\text{m}$ **without the use of additional optical**

components (e.g. GRIN lenses), a hollow ...

... as well as an arbitrary vector field on the HOPS with spatially variant localized elliptical polarizations (SLGM-5) (Fig. 1b). **Note that in this work, we did not include the phase profile of a lens in our metasurface design to collimate the beam, since our priority is to demonstrate the generation of complex beams using a fiber-integrated system. We should mention that a lens profile can be implemented onto our metasurface design when a collimated fiber output is required, which is empowered by the simultaneous and independent control of both the polarization and phase responses in our metasurface.** We show that 3D anisotropic meta-atoms with unleashed height degree of freedom offer a greatly...

R2.2: The meta atoms used are rectangular waveguides with width and length on the order of 0.5 to 1.8 micrometers. For these sub-micrometer waveguides, the eigen modes are not perfectly linear and could have significant cross-polarization component. Therefore the representation of the structured light field with Jones vectors on the linear polarization basis would have errors. Could the authors comment on accuracy of approximating the light field with by the linear Jones vectors?

We thank the Reviewer for the comment regarding the degree of linear polarization of the relevant modes formed in the meta-toms. To adequately address this point, we used Finite-Element simulations to calculate the two orthogonal modes in the 2D cross-sections of the meta-toms (Fig. R3, $n_{\text{polymer}} = 1.5$, $n_{\text{air}} = 1$, $W = 550$ nm, $L = 1.6$ μm , $P = 2.2$ μm , $\lambda = 1550$ nm). The results clearly show that the orientation of the electric field vectors is along either the x' or y' direction for virtually all points within the cross-section of the meta-atom, corresponding to a linear polarization for nearly the entire field. As expected, only very minor deviations are seen at the corners of the meta-atom, which are of no consequence and can be neglected. This minor deviation is expected and can be explained from a waveguide perspective by the small refractive index difference between core (polymer) and cladding (air). It should be noted that for larger refractive index differences (e.g. silicon to air), a stronger deviation from linear behavior can be expected. Overall, it can be stated that the modes in the polymeric meta-atoms used here are completely linearly polarized, allowing the use of the Jones vector formalism.

Fig. R3. Spatial distribution of the intensity (linear scale) and the orientation of the in-plane electric field (black arrows with normalized length) of the two relevant modes within the meta-atoms, simulated using finite element modelling. The left plot (a) shows the mode with a dominant polarization along the y' -direction, while the right plot (b) shows the orthogonal mode. The coordinate system ($x'y'$ -system) shown in the bottom left corners of the two plots refers to the orientation of the cross-section of the meta-atom and not to the laboratory system.

To account for the Reviewer's comment, the main text has been amended as follows: ... Eq. 2 can be used to define different vector beams on the HOPS with spatially variant polarization distributions, with some representative examples shown in Fig. 2a (from I to V). **Note that the two orthogonal modes formed in rectangular meta-atom cross-section are almost completely linearly polarized, justifying the use of the Jones-matrix formalism (see Supplementary Note 3 for further details.) ...**

Moreover, the simulation results including the discussion have been included into the Supplementary Information as follows: ...

...

Supplementary Note 3. Simulation of electric field intensity distribution inside nanopillar

To verify that the Jones formalism can be used in this work, Finite-Element simulations have been employed to calculate the two orthogonal modes in the 2D cross-sections of the meta-toms (Fig. S4, $n_{\text{polymer}} = 1.5$, $n_{\text{air}} = 1$, $W = 550$ nm, $L = 1.6$ μm , $P = 2.2$ μm , $\lambda = 1550$ nm). The results clearly show that the orientation of the electric field vectors is along either the x' or y' direction for virtually all points within the cross-section of the meta-atom, corresponding to a linear polarization for nearly the entire field. As expected, only very minor deviations are seen at the corners of the meta-atom, which are of no consequence and can be neglected. This minor deviation is expected and can be explained from a waveguide perspective by the small refractive index difference between core (polymer) and cladding (air). It should be noted that for larger refractive index differences (e.g. silicon to air), a stronger deviation from linear behaviour can be expected. Overall, it can be stated that the modes in the polymeric meta-atoms used here are completely linearly polarized, allowing the use of the Jones vector formalism.

Fig. S4. Spatial distribution of the intensity (linear scale) and the orientation of the in-plane electric field (black arrows with normalized length) of the two relevant modes within the meta-atoms, simulated using finite element modelling. The left plot (a) shows the mode with a dominant polarization along the y' -direction, while the right plot (b) shows the orthogonal mode. The coordinate system ($x'y'$ -system) shown in the bottom left corners of the two plots refers to the orientation of the cross-section of the meta-atom and not to the laboratory system.

...

R2.3: The experimentally measured intensity patterns of the SLGM outputs of the radial (Fig.3e) and azimuthal (Fig.3f) vectors deviate from the expected symmetric two-lobe patterns. What is the reason behind this, it is partly due to the approximation of the field vectors of the meta atoms by linear vectors? Or it is due to imperfection of the fabrication of the metal atoms? Some comments would help to reader to understand. I suggest to include the simulated results, compare with the measured ones and discuss the differences.

We thank the Reviewer for mentioning the intensity distributions in Fig. 3e and 3f. According to our simulations, we attribute the main reason for the deviation from the two-lobes pattern is due to the misalignment of the fiber output and the metasurface at the exit plane of a fiber tower. A tilt of the fiber end face with respect to the fiber axis leads to a slightly tilted beam (Fig. R4a), which could be caused by imperfect fiber cleaving technique. This misalignment mainly results from the fiber cleaving quality, with the ideal situation that the cutting cross-section is flat and perpendicular to the fiber axis. For practical fiber cleaving, however, there is some imperfection that leads to a shifted fiber beam output. To verify its impact, we simulated the influence of misalignment for the case of the sample producing azimuthal output polarization. Here we assume that the metasurface was misaligned by $20\mu\text{m}$ in x-direction and $5\mu\text{m}$ along the y-axis (after propagating $550\mu\text{m}$ from an imperfect cleaved fiber end face) with respect to the fiber output on the exit plane of the fiber tower. The impact of this misalignment was theoretically studied, and the results are shown in Fig. R4:

Fig. R4. Impact of misalignment between fiber output and metasurface on the imaging results, exemplified by considering an azimuthally polarized beam in the metasurface plane. (a) Schematic of misalignment due to the cleaved fiber end face not perpendicular to the fiber axis. **(b)** (left) Experimentally measured intensity patterns of the SLGM outputs of an azimuthal vector beam under different polarization filters. (right) Corresponding simulation which matches the experimental results. These calculations consider a misalignment between

fiber output and metasurface on the tower surface, with $20\mu\text{m}$ in x-direction and $5\mu\text{m}$ along the y-axis (after propagating $550\mu\text{m}$ from an imperfect cleaved fiber end face). Scale bar: $25\mu\text{m}$.

R2.4: The simulated and the measured intensity patterns in Fig.4 and 5 are also quite different. We thank the Reviewer for mentioning the differences of the intensity patterns of experiments and simulations. In accordance with our previous response, the main reason for this are fabrication inaccuracies that impact the nanopillars, which caused slight phase and transmission differences compared to simulations. More importantly, we find that for OAM-3 presented in Fig. 4, the differences of the intensity patterns of experiments and simulations are mainly due to imperfect fiber cleaving that induces a small tilt of the fiber end face with respect to the fiber axis and therefore, introducing a slightly misalignment between the fiber output and metasurface on the top of the hollow tower (Fig. R4a).

To account for the Reviewer's comment, we have modified the main text as follows:

...corroborating the nature of the transformed OAM beams (Fig. 4d). **The inhomogeneity of the intensity distribution in the experimental results is mainly due to misalignment between the fiber output beam and metasurface (Supplementary Note 11).** To consider only the OAM-induced beam divergence, we...

And we also added a Supplementary Note 11:

Supplementary Note 11. Impact of misalignment between metasurface and the fiber output.

According to our simulations, we attribute the main reason for the differences of the intensity patterns of structured light fields in experiments and simulations is due to the misalignment of the fiber output and the metasurface at the exit plane of a fiber tower. A tilt of the fiber end face with respect to the fiber axis leads to a slightly tilted beam (Fig.S12), which could be caused by imperfect fiber cleaving technique. This misalignment mainly results from the fiber cleaving quality, with the ideal situation that the cutting cross-section is flat and perpendicular to the fiber axis. For practical fiber cleaving, however, there is some imperfection that leads to a shifted fiber beam output. To verify its impact, we simulated the influence of misalignment for the case of the sample producing azimuthal output polarization. Here we assume that the metasurface was misaligned by $20\mu\text{m}$ in x-direction and $5\mu\text{m}$ along the y-axis (after propagating $550\mu\text{m}$ from an imperfect cleaved fiber end face) with respect to the fiber output on the exit plane of the fiber tower. Here we calculate the impact of this misalignment on the generated OAM modes with different topological charges of -1 and -3 (Fig. S12).

Fig. S12. Impact of misalignment between the fiber output and metasurface on the structured light transformation. (a) Schematic of misalignment resulting from the cleaved fiber end face not being perpendicular to the fiber axis. (b) Simulation results of the OAM beam output carrying $l=-1$, considering a $4\mu\text{m}$ shift along the x -direction (top), and output beam carrying $l=-3$ with a $7\mu\text{m}$ shift along the y -axis in the metasurface plane (bottom). The black and purple dots and the corresponding dotted lines represent the beam and metasurface, respectively. (c) Simulation results in Fourier Plane with the incident beam in (b). (d) The experiment results in our SLGM-3 and 4. The dashed circles in (c) mark the beam divergence angle of 0.06 and 0.1, respectively.

R2.5: It would be nice to state at least one specific application of the device.

We thank the Reviewer for the suggestion. We identified several promising application fields of our structured light metafibers, which was already mentioned in the submitted version of the manuscript: “Our work provides a new paradigm for advancing optical fiber science and technology towards fiber-integrated light shaping, which may find important applications in fiber communications, fiber lasers and sensors, endoscopic imaging, fiber lithography, and lab-on-fiber technology.” Here we articulate one specific example of using our metafibers for structured light imaging and microscopy as follows.

Structured light imaging and microscopy: Structured light has found profound impact on both exploiting new phenomena in light-matter interactions [Litchinitser 2012] and offering a new multiplexing scheme for optical imaging and data storage [D'Ambrosio et al. 2015]. Recently, helical beams carrying orbital angular momentum have shown efficient chiral sensitivity [Bégin, Jain et al. 2022; Ni, Liu et al. 2021], which may open new opportunities in chiroptical spectroscopy. In a conventional structured light microscope, to prepare different structured light fields, a set of bulky and costly optical components, such as waveplates, lenses, and spatial light modulators must be cascaded on a table-based system. This significantly increases the system’s complexity and cost. On the other hand, our structured light metafibers can directly generate arbitrary structured light fields on the hybrid-order Poincaré sphere, reducing the system complexity and cost. More importantly, owing to the use of single-mode

fibers, our transformed arbitrary structured light modes can be easily delivered to a distant end without suffering from transmission or bending loss, mode distortion, and beam divergence. Therefore, we believe the on-demand generation of arbitrary structured light modes and the “plug and play” feature of our metafibers could add significant benefits to optical microscopes.

To account for the Reviewer’s comment, we have modified the text as follows: ... technology towards multimode light shaping and multi-dimensional light-matter interactions. **Structured light has found profound impact on both exploiting new phenomena in light-matter interactions⁷⁴ and offering a new multiplexing scheme for optical imaging and data storage⁷⁵. Recently, helical beams carrying orbital angular momentum have shown efficient chiral sensitivity^{76,77}, which may open new opportunities in chiroptical spectroscopy. In a conventional structured light microscope, to prepare different structured light modes, a set of bulky and costly optical components, such as waveplates, lenses, and spatial light modulators must be cascaded on a table-based system. This significantly increases the system’s complexity and cost. On the other hand, our structured light metafibers can directly generate arbitrary structured light fields on the hybrid-order Poincaré sphere, reducing the system complexity and cost. More importantly, owing to the use of single-mode fibers, our transformed arbitrary structured light modes can be easily delivered to a distant end without suffering from transmission or bending loss, mode distortion, and beam divergence. Therefore, we believe the on-demand generation of arbitrary structured light fields and the “plug and play” feature of our metafibers could add significant benefits to optical microscopes. For instance, long-distance transmission and delivery of structured ...**

[74] N. M. Litchinitser, Structured light meets structured matter. *Science* 337, 1054-1055 (2012).

[75] V. Parigi, V. D'Ambrosio, C. Arnold, L. Marrucci, F. Sciarrino, J. Laurat, Storage and retrieval of vector beams of light in a multiple-degree-of-freedom quantum memory. *Nat Commun* 6, 7706 (2015).

[76] J.-L. Bégin et al., Nonlinear helical dichroism in chiral and achiral molecules. *Nature Photonics* 17, 82-88 (2022).

[77] J. Ni et al., Giant Helical Dichroism of Single Chiral Nanostructures with Photonic Orbital Angular Momentum. *ACS Nano* 15, 2893-2900 (2021).

R2.6: In Supplementary Note 1, with the authors' definition of matrix M , $E_{out} = RMR^{-1} E_{in}$ seems incorrect. R and R^{-1} are already included in M ?

We would like to apologize for our error in deriving the equation. To address the Reviewer's comment, we have modified the formula in Supplementary Note 4 as follows: ... Thus, for an arbitrary input light with a characteristic electric field E_{in} passing through the meta-atoms, the output light electric field can be described by $E_{out} = ME_{in}$

Reviewer #3 (Remarks to the Author)

The manuscript titled "Metafiber: Transforming Arbitrarily Structured Light" presents a novel metafiber platform capable of generating arbitrarily structured light output on the HOPS through the use of nanoprinted metasurfaces. The researchers employed polymeric metasurfaces and laser nanoprinting techniques to achieve this breakthrough and successfully interfaced it with single-mode fibers. This platform represents a significant advancement over traditional optical fiber technology and holds enormous potential for various applications. While the results are impressive, there remain several important and unresolved issues that require further exploration.

We appreciate the Reviewer's thorough reading and valuable comments that have contributed to manuscript enhancement. We have provided point-to-point responses below to address each comment and outline the corresponding refinements made based on the Reviewer's feedback.

Here are the detailed comments:

R3.1: There have been related articles to implement OAM transmission on optical fiber, such as *Laser & Photonics Reviews* 15.5 (2021): 2000581. *Optica* 9.6 (2022): 645-651. *Appl. Sci.* 2019, 9(5), 1033, so the expression of "To date, the generation of structured light is still handled outside the fiber via bulky optics in free space" should be modified more exactly.

We thank the Reviewer for pointing out the inaccurate description. We reviewed these articles and modified the text in the manuscript as follows:

... To date, the generation of structured light is handled **in the majority of cases** outside the fiber via bulky optics in free space, which **can** hinder the deployment of structured light for fiber science and technology and partially nullifies the advantages of optical fiber such as flexible light guidance. **Here, promising approaches implementing meta-structures on fiber end faces have been used to demonstrate Bessel beam converters⁴⁴, fiber couplers⁴⁵, polarization controllers and waveplates⁴⁶. ...**

[44] I. V. A. K. Reddy, A. Bertocini, C. Liberale, 3D-printed fiber-based zeroth- and high-order Bessel beam generator. *Optica* 9, (2022).

[45] C. Zhou et al., All-Dielectric Fiber Meta-Tip Enabling Vortex Generation and Beam Collimation for Optical Interconnect. *Laser & Photonics Reviews* 15, (2021).

[46] H. Zhang, B. Mao, Y. Han, Z. Wang, Y. Yue, Y. Liu, Generation of Orbital Angular Momentum Modes Using Fiber Systems. *Applied Sciences* 9, (2019).

R3.2: The aspect ratio of the nanopillars is very high, so, as we all know, the antenna may collapse very easily. Using 3D printing technology, how accurate is the structure, and how about the morphology and stability of the structure?

We thank the Reviewer for raising the question related to the accuracy of the nanopillars.

Prevention of collapsing: As the Reviewer mentioned, the high aspect ratio pillars are easy to fall over, which is the main challenge in fabrication. Here, we used two strategies to prevent nanopillars from collapsing. The first strategy is to choose suitable nanopillars when designing

the metasurface. In more detail, choose nanopillars with a suitable length/period ratio to avoid falling over. The second strategy is optimizing the printing parameters to enhance mechanical stability via improving the degree of polymerization through optimizing the hatching (transverse plane) and slicing (height dimension) distances. Specifically, the falling over of the nanopillars has been circumvented through employing an optimization procedure of the nanoprinting recipe mainly depending on three parameters: laser power (exposure energy), scanning speed (exposure time) and scanning resolution (slicing and hatching distance), all of which are related to the polymerization degree of the photoresist used during printing. A higher level of polymerization provides higher mechanical strength, resulting in high aspect ratios between 16 and 24 of our nanopillars. However, excessive laser energy can cause overexposure of the photoresist, while slow scanning speed may impose the entire printing process to be time-consuming. To take all these issues into account, we optimized the recipe for nanoprinting to find a balance between the realization of mechanically stable nanopillar structures and a reasonable printing time.

To account for the Reviewer's comment, we have modified the text as follows: ...out of the fiber has a much smaller diameter at the location of the metasurface (100 μ m). **To increase the mechanical stability of the high-aspect ratio polymer nanopillars, we have used the following optimized printing parameters: laser power 55mW, scanning speed 3500 μ m/s, and small hatching (lateral laser movement step: 10nm) and slicing (axial laser movement step: 20nm) distances.** The printing process takes a total of 10-13 hours per sample, including 5 hours for the 3D hollow...

Accuracy of the nanoprinted structure: Due to the photopolymerization process, the fabricated structures are generally different from the designed ones. To make the metasurface more accurate, we first calibrated the geometric size of nanopillars. Here we fabricated a set of identical nanopillars with different sizes and tested the repeatability and we found that the fabrication uncertainty (due to machine error) is as small as 15nm.

To illustrate the accuracy of fabrication here, we added a Supplementary Note 6 in which we discuss the calibration and the precision that is reachable in fabrication:

Supplementary Note 6. Size calibration of 3D laser-nanoprinted nanopillars.

Due to the photopolymerization process, the fabricated structures are generally different from the designed ones. To make the metasurface more accurate, we first calibrated the geometric size of nanopillars. Here we fabricated a set of identical nanopillars with different sizes and tested the repeatability and we found that the fabrication uncertainty (due to machine error) in the lateral directions is as small as 15nm.

Lateral size calibration

In lateral size calibration, the calibration was done for nanopillar length and width dimensions. In horizontal direction, we design different nanopillars with length changing from 600nm to 1200nm with a step of 100nm. We measured the length of pillars under SEM and obtained the relationship between the design and measured results by fitting (Fig S6).

Fig. S6. Calibration of the length size of 3D laser-nanoprinted nanopillars. The images on top are the SEM image of nanopillars with different lengths (scale bar: 500nm). Left to right: nanopillars with designed lengths of 600nm, 800nm, 1000nm and 1200nm. The black solid line is the fitted relationship between designed length and the measured length.

There is a linear relationship between the design and the actual printing size. So, the fitting relationship is:

$$L_{measured} = 1.09 * L_{designed} + 205.2 \text{ nm}$$

The same method is used for the width characterization and the experiment results are shown in Fig S7. Similarly, the relationship between design and actual printed size can be written as:

$$W_{measured} = 1.03 * W_{designed} + 341.2 \text{ nm}$$

The offset values in the lateral directions are determined from the diffraction-limited focal spot of a tightly focused linearly polarized light.

Fig. S7. Calibration of the width size of 3D laser-nanoprinted nanopillars. The images on top are the SEM image of nanopillars with different widths (scale bar: 500nm). Left to right: nanopillars with designed widths of 100nm, 200nm, 300nm and 400nm. The black solid line is the fitted relationship between designed width and the measured width.

Long-term stability: As for the long-term stability, the implemented SLGMs still show no signs of performance deterioration after 6 months of storage in air. In addition, we would like to mention that we have produced light-guiding structures with much higher aspect ratios of cylindrical strands (so-called hollow-core light cages [Bürger, Kim et al. 2021]), which also showed no signs of degradation over many months. For example, these light cages were employed in quantum optical alkali vapor experiments and used for measurements over many months [Davidson-Marquis, Gargiulo et al. 2021]. No signs of degradation of the optical properties were measured, which is remarkable since the light cages were exposed to a chemically aggressive environment.

To account for the Reviewer's comment, we have modified the text as follows: ... by immersing them in propylene glycol monomethyl ether acetate (PGMEA, Sigma-Aldrich) for 20 min, Isopropanol (IPA, Sigma-Aldrich) for 5 min, and Methoxy-nonafluorobutane (Novec 7100 Engineered Fluid, 3M) for 2 min, respectively. **The nanopillars reveal a mechanical long-term stability, since the implemented SLGMs show no signs of performance deterioration after 6 months of storage in air. This behavior is in agreement with our experience with strand-based light-guiding structures (so-called light cages^{78, 79}) that include polymer cylinders of even higher aspect ratios. ...**

[78] J. Bürger, J. Kim, B. Jang, J. Gargiulo, M. A. Schmidt, S. A. Maier, Ultrahigh-aspect-ratio light cages: fabrication limits and tolerances of free-standing 3D nanoprinted waveguides. *Optical Materials Express* 11, (2021).

[79] F. Davidson-Marquis et al., Coherent interaction of atoms with a beam of light confined in a light cage. *Light Sci Appl* 10, 114 (2021).

R3.3: In the paper, the height of 3D nanopillars are unlocked in one metasurface. The phase response of the nanopillar is related to the height. Whether crosstalk exists in the phase response between adjacent nanorods of different heights in the same structure? What are the differences and advantages between the same height nanopillars and the different height nanopillars in one metasurface?

We thank the Reviewer for this comment. Before going into details, we would like to mention that the phase of transmitted light from the nanopillars does not depend exclusively on the height of the pillars. Rather, there is an interplay between the different geometric parameters along as well as perpendicular to the axis of the nanopillars, allowing flexible control of the phase. Thus, changing the height of the nanopillars represents one, but not the only, degree of freedom to control the phase of the elementary waves.

Crosstalk: Here we would like to mention that crosstalk in the studied structures results exclusively from the interaction of the modes of neighboring cylinders. Here, only the distribution of the modes in the plane perpendicular to the cylinders plays a role, which is independent of the height of the cylinders. We are currently preparing a manuscript that examines metasurfaces from the point of view of modes within cylinder arrays and thus addresses precisely the issue discussed here. The main conclusion of this study is that the accumulated phase, which is formed during propagation through each cylinder, is essentially determined by the phase index of the lattice mode. The mode structure, which arises in the plane perpendicular to the cylinder axis, plays the central role. Thus, crosstalk is present, which is relevant but accounted for in the simulations by the periodic boundary conditions.

Advantage of the height degree-of-freedom: For a waveguide-based metasurface, it is well known that the radiated phase of each element is essentially the product of the modal phase index n_{eff} and the propagation length, i.e., cylinder height h . Here, the nanoprinting process uniquely allows the height of the individual elements to be adjusted. This provides another degree of freedom not available in metasurfaces with constant height elements, ultimately providing greater flexibility in the design of the anticipated optical functionality. We would like to mention here that this degree of freedom was exploited in our previous works to achieve both complex-amplitude and achromatic operations of metasurfaces. As shown in these works, this functionality cannot in principle be achieved with constant height elements. For further details, we would like to refer the attention of the Reviewer to the discussion in the context of Fig. 3 of our previously published works [Ren, Jang et al. 2022, Ren, Fang et al. 2020].

To account for the Reviewer's question, we have modified the main text of the manuscript as follows: ...Therefore, our results suggest that 3D nanopillars with the unlocked height degree of freedom provide a powerful platform for implementing arbitrarily structured light. **Note that the ability to assign an individual height to each nanopillar is one of the key advantages of the nanoprinting process over other planar lithography methods, which typically yield nanostructures of identical height.**^{10,60} (Details in Supplementary Note 5) ...

[10] H. Ren, X. Fang, J. Jang, J. Burger, J. Rho, S. A. Maier, Complex-amplitude metasurface-based orbital angular momentum holography in momentum space. *Nat Nanotechnol* **15**, 948-955 (2020).

[60] H. Ren et al., An achromatic metafiber for focusing and imaging across the entire telecommunication range. *Nat Commun* **13**, 4183 (2022).

R3.4: Time spent is an important indicator of 3D printing. How long does it take to print the whole structure?

We thank the Reviewer for pointing out the duration required for manufacturing, which is essential for potential applications. The realization of the tower takes 5 hours and that of the metasurface 5–8 hours depending on the design. Thus, the duration of the manufacturing process of the entire nanoprinted structure takes a total of 10–13 hours per sample.

To account for the Reviewer's comment, we have modified the main text as follows: ...out of the fiber has a much smaller diameter at the location of the metasurface (100 μm). **To increase the mechanical stability of the high-aspect ratio polymer nanopillars, we have used the following optimized printing parameters: laser power 55mW, scanning speed 3500 $\mu\text{m/s}$, and small hatching (lateral laser movement step: 10nm) and slicing (axial laser movement step: 20nm) distances. The nanoprinting process takes a total of 10–13 hours per sample, including 5 hours for the hollow tower and 5–8 hours for the metasurface depending on the design.** After laser exposure, the samples were developed by immersing...

R3.5: Why Eq(3) indicates that both the polarization and phase of an output beam can be controlled simultaneously by a single nanopillar, please give further detailed derivation.

We thank the Reviewer for the comments on the operational principle of nanopillars-based metasurfaces that are designed based on Eq. 3, which can actually be divided into two parts:

The first part $e^{i\varphi_x}$ refers to the phase of the output light along the x-axis. Since the nanopillars can be regarded as a truncate waveguide, their scattered phase $e^{i\varphi_x}$ is mainly determined by the elements' height and transverse dimensions.

The second part $\begin{bmatrix} t_x \cos^2(\gamma) + t_y \sin^2(\gamma) e^{i\Delta\varphi} \\ \frac{1}{2}(t_x - t_y) e^{i\Delta\varphi} \sin(2\gamma) \end{bmatrix}$ controls the polarization, i.e., the Jones vector.

The rectangular shape makes the nanopillars birefringent which appears in this equation as the phase retardation $\Delta\varphi$. When we change the size of nanopillar, its phase and phase retardance responses change at the same time. Owing to the unleashed height degree of freedom, our 3D meta-atom library provides us with a complete and precise selection of 3D nanopillars with desired phase and phase retardance responses. At the same time, the in-plane rotation angle γ operating as an independent parameter can contribute to the control of the outgoing polarization. Therefore, both the polarization and phase of the transmitted light from nanopillars can be controlled simultaneously by adjusting the dimensions and orientation of a single nanopillar.

To explain the equation clearer, we have modified the main text as follows: ... **All these parameters are spatially dependent as functions of x and y. Equation 3 indicates that both the polarization (controlled by $\Delta\varphi$ and γ) and phase (φ_x) of an output beam can be controlled simultaneously by a single nanopillar, which forms the physical basis for implementing any arbitrarily structured light fields. Note that φ_x controls the transmitted phase from each nanopillar waveguide, while $\Delta\varphi$, because of the rectangular cross-section of the element, induces modal birefringence, together with the in-plane rotation angle γ allow**

the control of the polarization of transmitted light. The key to our design is to find 3D nanopillars to be arranged in a particular...

R3.6: How much power the structure can withstand and how efficient the structure is to light conversion, please give further supplements.

We thank the Reviewer for the comment, which we address in the following:

Power stability: The power stability of nanoprinted structures on fibers has been discussed in detail in one of our previous works [Ren, Jang et al. 2022]. The key point is the comparatively large transverse extent of the beam at the location of the metasurface (on the order of 100 μm), which results in sufficiently low local intensity at the location of the metasurface and the individual element to prevent damage to the nanoprinted structure. Here, we would like to refer to the discussion shown in Supplementary Note 9 and Fig. S17 of Ref 60. In that work, the power stability is discussed in terms of (i) literature research, (ii) measurements of the power stability using a test sample, and (iii) previous experiments that use nanoprinted structures on fibers for optical trapping. All three aspects confirm that the nanoprinted structures remain unaffected at the level of light powers involved in the project.

To consider the Reviewer's comment, the main text has been changed as follows: ... Specifically, the lobes follow the orientation of the polarization axis of the linear polarizer, confirming the generation of cylindrically polarized beams. **Note that due to the relatively large extension of the optical beam at the location of the metasurface, the local light intensity is sufficiently low to prevent damaging of the sample. Further details can be found in Supplementary Note 9 and Fig. S17 of Ref. 60. ...**

[60] H. Ren et al., An achromatic metafiber for focusing and imaging across the entire telecommunication range, *Nat Commun.* 13, 4183 (2022).

Conversion efficiency: Our metasurface consists of many subwavelength pixels that can be regarded as waveplates converting incident linear polarization into spatially variant polarization states on a vector beam. Since we have demonstrated many different polarization outputs including linear, circular, and elliptical polarizations, to simplify the verification process, here we fabricated a half wave-plate-type metasurface on glass and measured its conversion efficiency of 67%. This represents the case that our metasurface can convert the left-handed circularly polarization (LCP) into right-handed circularly polarization (RCP), and vice versa.

To consider the Reviewer's comment, the main text has been changed and a **Supplementary Note 7 was added:**

...behaves like a half-wave plate **with a measured conversion efficiency of around 67%.** (**Supplementary Note 7**) The nanopillar waveguide has ...

Supplementary Note 7. Conversion efficiency of a half-wave-plate metasurface sample.

Our metasurface consists of many subwavelength pixels that can be regarded as waveplates converting incident linear polarization into spatially variant polarization

states on a vector beam. Since we have demonstrated many different polarization outputs including linear, circular, and elliptical polarizations, to simplify the verification process, here we fabricated a half wave-plate-type metasurface on glass and measured its conversion efficiency of 67%. This represents the case that our metasurface can convert the left-handed circularly polarization (LCP) into right-handed circularly polarization (RCP), and vice versa. Our measurement setup is shown in Fig S8. The conversion efficiency is defined as the intensity ratio of the cross-polarization component normalized to the incident light, e.g., I_{RCP}/I_{LCP} .

Fig. S8. (a) SEM image of a half-wave-plate-type nanopillar metasurface (scale bar: 20 μ m). (b) Optical setup for measuring polarization conversion efficiency of the metasurface on a glass substrate.

REVIEWERS' COMMENTS

Reviewer #1 (Remarks to the Author):

The research work reported in the manuscript is very interesting. The author has revised the paper according to the reviewer's comments and has also answered the reviewer's questions. I recommend publishing this paper.

Reviewer #2 (Remarks to the Author):

This paper reports the integration of meta-surfaces onto the end-face of optical fibers for transforming light field. The results are interesting and could have applications in structured light systems with optical fibers.

The issues proposed in my last review are addressed sufficiently.

Reviewer #3 (Remarks to the Author):

The paper has been revised and improved according to the reviewer's comments. The quality of the paper has been significantly enhanced, and it is now considered suitable for acceptance.

Dear Editor,

Thank you for letting us know the feedback of Reviewers on our manuscript titled "Metafiber transforming arbitrarily structured light". We are delighted to know that our previous revision has satisfied all the Reviewers.

Reviewer #1 (Remarks to the Author):

The research work reported in the manuscript is very interesting. The author has revised the paper according to the reviewer's comments and has also answered the reviewer's questions. I recommend publishing this paper.

Reviewer #2 (Remarks to the Author):

This paper reports the integration of meta-surfaces onto the end-face of optical fibers for transforming light field. The results are interesting and could have applications in structured light systems with optical fibers.

The issues proposed in my last review are addressed sufficiently.

Reviewer #3 (Remarks to the Author):

The paper has been revised and improved according to the reviewer's comments. The quality of the paper has been significantly enhanced, and it is now considered suitable for acceptance.